# Set-Coupled Guidance: Set-Level Coordination in Diffusion-Based Dataset Distillation

**Ziang Gan**[1]   **Qi Zhu**[1]   **Libao Zhang**[1]

## Abstract

Diffusion models serve as generative priors for dataset distillation, yet existing pipelines rely on per-sample update rules that evolve each synthetic image independently, limiting their ability to optimize collective set-level objectives. We propose Set-Coupled Guidance (SCG), a plug-and-play auxiliary controller that shifts from per-image to group (IPC-at-once) sampling by injecting set-symmetric feedback at each diffusion step. SCG combines spectral set-point regulation, which aligns set-level statistics to real data via empirical characteristic function matching, with cooperative kernel coupling that stabilizes joint trajectories under noisy feedback. All computations operate on lightweight descriptors extracted from predicted clean latents, adding low overhead to the base method. We provide theoretical analysis including Lyapunov descent and input-to-state stability for distributional tracking. Experiments on ImageNette, ImageWoof, ImageNet-100 and ImageNet-1K show consistent accuracy gains across multiple diffusion-based baselines; code is available at https://github.com/tadels/SCG.

## 1. Introduction

Dataset distillation (DD) (Wang et al., 2018) aims to synthesize a compact set $\mathcal{S}$ such that models trained on $\mathcal{S}$ achieve comparable performance to those trained on the full dataset $\mathcal{T}$. Such compact representations enable continual learning (Wiewel & Yang, 2021), accelerate neural architecture search (Such et al., 2020), and support privacy-preserving learning (Dong et al., 2022). Classical methods directly optimize synthetic pixels by matching some proxy of dynamics, including gradient matching (Zhao et al., 2021), distribution matching (Zhao & Bilen, 2023; Wang et al., 2022), and trajectory matching (Cazenavette et al., 2022; Du et al., 2023). However, pixel-level optimization faces limitations at scale: memory costs grow prohibitively with resolution (Yin & Shen, 2024; Cui et al., 2023), synthesized images can exhibit visual artifacts (Yu et al., 2023; Cazenavette et al., 2023), and reliance on proxy models (e.g., for gradient or trajectory matching) introduces architectural bias, causing the resulting data to transfer poorly across architectures (Yu et al., 2023). Diffusion models (Ho et al., 2020; Rombach et al., 2022) have been adopted as generative priors for dataset distillation (Su et al., 2024; Santiago et al., 2025; Chen et al., 2025; Cui et al., 2025; Zhao et al., 2025; Wang et al., 2025a; Moser et al., 2025), which constrain samples to manifolds and enable higher-resolution distillation with cross-architecture transfer.

Many diffusion-based DD methods leverage pre-trained diffusion models to generate distilled data. One line of work trains or fine-tunes the generative model itself, e.g., Minimax Diffusion (Gu et al., 2024) and DiM (Wang et al., 2024). Among methods that keep the diffusion model frozen, some inject guidance gradients into the reverse process, e.g., $D^4M$ (Su et al., 2024), $MGD^3$ (Santiago et al., 2025), IGD (Chen et al., 2025), and OT-GDD (Cui et al., 2025), while others optimize static latent codes, e.g., $CaO_2$ (Wang et al., 2025a) and LD3M (Moser et al., 2025). Despite these differences, many frozen-backbone methods provide a *per-sample* update direction that depends on the sample's current state, leading to *separable* within-class dynamics in which samples evolve independently.

This separability is limiting. The distilled data for each class is a *per-class synthetic set*, typically reported as images per class (IPC); what matters is not individual image quality but the *joint configuration* of the IPC samples and its measurable set-level statistics (e.g., coverage, diversity, distributional fidelity). Separable dynamics cannot encode objectives that depend on the joint within-class state, as each sample evolves independently of its peers, leaving per-sample objectives with no mechanism to detect or correct set-level redundancy. Concretely, objectives such as diversity/coverage and distributional matching are non-additive over IPC samples, so purely per-sample guidance cannot

---

[1]School of Artificial Intelligence, Beijing Normal University, Beijing 100875, China. Correspondence to: Libao Zhang <libaozhang@bnu.edu.cn>.

*Proceedings of the 43$^{rd}$ International Conference on Machine Learning*, Seoul, South Korea. PMLR 306, 2026. Copyright 2026 by the author(s).

correct set-level redundancy on-the-fly.

Several methods attempt to address this limitation. D³HR (Zhao et al., 2025) generates multiple candidate subsets by varying the initial latents and selects the best one post-hoc; it does not inject explicit guidance into the reverse diffusion trajectory, and samples do not interact during generation. IGD (Chen et al., 2025) introduces sequential conditioning that forms a causal (lower-triangular) coupling where each sample depends only on those generated before it. OT-GDD (Cui et al., 2025) uses optimal transport to match a synthetic batch to a real batch (via an OT plan), guiding reverse diffusion toward batch-wise alignment, yet the coupling remains indirect (mediated by batch-to-data matching rather than symmetric IPC-to-IPC feedback). Neither achieves the dense, symmetric coordination needed to optimize set-level statistics during generation. In contrast, our controller imposes a fully symmetric coupling in which every sample's update depends on the entire within-class state simultaneously.

We propose *Set-Coupled Guidance* (SCG), a plug-and-play auxiliary controller that shifts from per-image sampling to *group (IPC-at-once) sampling*. At each update step, SCG injects a set-symmetric controller $u_n^{(k)}(\mathcal{Z}_n^c, t_n)$ that explicitly depends on the *entire* within-class state $\mathcal{Z}_n^c$, enabling dense coupling without modifying the base method or re-training the diffusion model. Two complementary components drive this coupling: *spectral set-point regulation* aligns set-level statistics to real data by matching empirical characteristic function (ECF) features, while *cooperative kernel coupling* regularizes joint trajectories for robustness under noisy feedback. Crucially, all computations operate on lightweight descriptors extracted from predicted clean latents, avoiding backpropagation through the diffusion model. We provide theoretical guarantees including Lyapunov-based convergence, input-to-state stability (ISS) for set-point tracking, and an anti-collapse property when real data is non-degenerate.

**Contributions.**

- We propose *Set-Coupled Guidance* (SCG), a plug-and-play controller that enables dense within-class coupling via spectral regulation and kernel-based stabilization, compatible with both guided sampling and latent refinement pipelines.

- We provide theoretical analysis establishing Lyapunov descent, ISS tracking of distributional statistics, and an anti-collapse property when real data is non-degenerate.

- We demonstrate consistent improvements over multiple base methods across ImageNette, ImageWoof,

ImageNet-100 and ImageNet-1K, with negligible computational overhead.

## 2. Related Work

### 2.1. Dataset Distillation.

Dataset distillation (DD) synthesizes a compact labeled set $\mathcal{S}$ such that models trained on $\mathcal{S}$ achieve comparable performance to those trained on the full dataset $\mathcal{T}$ (Wang et al., 2018). DD has been applied to continual learning (Wiewel & Yang, 2021), neural architecture search (Such et al., 2020), and privacy-preserving scenarios (Dong et al., 2022). Classical methods directly optimize synthetic pixels by matching some proxy of training dynamics, including gradient matching (Zhao et al., 2021), distribution matching (Zhao & Bilen, 2023; Zhao et al., 2023; Wang et al., 2022), trajectory matching (Cazenavette et al., 2022; Du et al., 2023; Guo et al., 2023), representative matching (Liu et al., 2023), and kernel inducing points (Nguyen et al., 2021; 2020). Feature-regression and memory-based methods (Zhou et al., 2022; Deng & Russakovsky, 2022) reduce storage via compact representations. Recent large-scale and distributional variants further refine non-critical image regions (Tran et al., 2025), use neural characteristic-function matching (Wang et al., 2025b), optimize Wasserstein-metric distribution matching (Liu et al., 2025), or synthesize multimodal data from image–text prototypes (Choi et al., 2026). However, pixel-level optimization faces limitations at scale: high memory cost (Yin & Shen, 2024; Cui et al., 2023), synthesized images can show artifacts (Yu et al., 2023; Cazenavette et al., 2023), and poor cross-architecture transfer (Yu et al., 2023), motivating the use of generative priors.

### 2.2. Dataset Distillation with Diffusion Models.

Diffusion models generate data by iteratively denoising a latent variable using a learned noise estimator (Ho et al., 2020), and are often scaled using latent diffusion (Rombach et al., 2022). Diffusion-based DD leverages these models as generative priors to produce architecture-agnostic distilled data at higher resolutions, and is compared against large-scale baselines such as SRe²L (Yin & Shen, 2024) and RDED (Sun et al., 2024).

Existing diffusion-based DD methods differ in their generation or optimization strategies. Guided sampling or inversion-based methods modify the reverse process, e.g., D⁴M (Su et al., 2024), MGD³ (Santiago et al., 2025), IGD (Chen et al., 2025), OT-GDD (Cui et al., 2025), and DAP (Su et al., 2026); latent selection/refinement methods operate on static latents, e.g., D³HR (Zhao et al., 2025), CaO₂ (Wang et al., 2025a) and LD3M (Moser et al., 2025); and generator-centric approaches train or fine-tune the generative model itself, e.g., Minimax Diffusion (Gu et al.,

2024) and DiM (Wang et al., 2024). Despite these differences, most pipelines still update samples with per-sample directions, leaving set-level statistics unoptimized. Outside diffusion-based DD, MIM4DD (Shang et al., 2023) encourages within-class diversity via mutual information. Our method introduces a plug-and-play controller that couples sets at each step via spectral regulation and kernel coupling. Concretely, SCG injects set-symmetric feedback from descriptors, enabling coordination without backpropagating through the diffusion.

## 3. Preliminaries

### 3.1. Dataset Distillation

Let $\mathcal{T} = \{(x_i, y_i)\}_{i=1}^{N_\mathcal{T}}$ denote a training set where $x_i \in \mathcal{X}$ are inputs and $y_i \in \mathcal{Y}$ are labels. Dataset distillation constructs a synthetic set $\mathcal{S} = \{(\tilde{x}_j, \tilde{y}_j)\}_{j=1}^{K}$ with $K \ll N_\mathcal{T}$, such that a model $f_\theta : \mathcal{X} \to \mathcal{Y}$ trained on $\mathcal{S}$ generalizes comparably to one trained on $\mathcal{T}$ (Wang et al., 2018). Let $\mathcal{L}(\theta; \mathcal{D}) = \frac{1}{|\mathcal{D}|} \sum_{(x,y) \in \mathcal{D}} \ell(f_\theta(x), y)$ denote the empirical risk with per-sample loss $\ell$. The optimization objective is to minimize the validation loss on $\mathcal{T}$:

$$\min_{\mathcal{S}} \mathcal{L}(\theta^\star(\mathcal{S}); \mathcal{T}), \tag{1}$$

where $\theta^\star(\mathcal{S}) = \arg\min_\theta \mathcal{L}(\theta; \mathcal{S})$ denotes the model parameters obtained by training on $\mathcal{S}$.

### 3.2. Diffusion Models

Let $x_0 \in \mathbb{R}^d$ be a data sample (pixel or latent space). The forward process corrupts $x_0$ into $x_t$ via:

$$x_t = \sqrt{\bar{\alpha}_t}\, x_0 + \sqrt{1 - \bar{\alpha}_t}\, \epsilon, \tag{2}$$

where $\epsilon \sim \mathcal{N}(0, I)$ and $\bar{\alpha}_t = \prod_{s=1}^{t}(1 - \beta_s)$ with noise schedule $\{\beta_t\}_{t=1}^{T}$. DDPM learns a noise predictor $\epsilon_\theta(x_t, t)$ and generates samples via the reverse iteration (Ho et al., 2020):

$$x_{t-1} = \frac{1}{\sqrt{\alpha_t}}\Big(x_t - \frac{\beta_t}{\sqrt{1 - \bar{\alpha}_t}}\epsilon_\theta(x_t, t)\Big) + \sigma_t \xi, \tag{3}$$

where $\xi \sim \mathcal{N}(0, I)$, $\alpha_t = 1 - \beta_t$, and $\sigma_t$ follows the variance schedule (e.g., $\sigma_t = \sqrt{\beta_t}$). This discrete process can be viewed as a discretization of a stochastic differential equation (Song et al., 2021). Guided generation modifies (3) by adding a gradient term $\nabla_{x_t} G(x_t)$ to steer sampling toward high-$G$ regions (Dhariwal & Nichol, 2021).

## 4. Method

Diffusion-based DD methods typically generate each distilled image via guided sampling (Santiago et al., 2025) or refine static latent codes (Wang et al., 2025a) independently,

producing a per-sample update direction that depends only on the individual state. However, for each class we distill a set of synthetic examples; what matters is the joint configuration of the set and its measurable set-level statistics.

To systematically characterize this limitation, we unify existing methods into a discrete-time state-space representation. Let $z_n^{(k)} \in \mathbb{R}^d$ denote the latent state of sample $k$ at iteration $n$, with associated diffusion timestep $t_n$. A single update step can be written as:

$$z_{n+1}^{(k)} = f(z_n^{(k)}, t_n) + \sigma(t_n)\, \xi_n^{(k)} + B(t_n)\, g^{(k)}(z_n^{(k)}, t_n), \tag{4}$$

where $f(z, t) + \sigma(t)\xi$ corresponds to the DDPM reverse step (3), $B(t_n)$ is the input gain, $g^{(k)}$ is the per-sample guidance, and $\xi_n^{(k)} \sim \mathcal{N}(0, I)$. For latent refinement, e.g., CaO$_2$ (Wang et al., 2025a), $f(z, t) = z$, $B(t_n)$ is the step size, $\sigma(t_n) = 0$, and $t_n$ is sampled per iteration to form a stochastic gradient estimate. Detailed instantiations for specific methods are provided in Section A.1.

Within this framework, the limitation becomes explicit: $g^{(k)}$ depends only on the individual state $z_n^{(k)}$, so the base dynamics are separable across the $K_c$ samples and cannot encode objectives that depend on the joint within-class state $\mathcal{Z}_n^c$. Our key idea is to shift from per-image sampling to *group (IPC-at-once) sampling* by injecting a set-symmetric controller that couples all within-class samples at each step. An overview is shown in Figure 1.

### 4.1. Overall Architecture

Our goal is to inject a set-level coupling term that explicitly depends on the *entire* within-class configuration, while preserving the base method's update logic and avoiding any retraining of the diffusion model.

To this end, we augment the base update with an auxiliary controller $u_n^{(k)}$. For a class $c$ with $K_c$ samples, let $\mathcal{Z}_n^c = (z_n^{(1)}, \ldots, z_n^{(K_c)})$ denote the joint latent state. In this work we use class-balanced distillation, so $K_c = \text{IPC}$ is a class-independent constant. The augmented dynamics become:

$$\begin{aligned} z_{n+1}^{(k)} = &\, f(z_n^{(k)}, t_n) + \sigma(t_n)\, \xi_n^{(k)} \\ &+ B(t_n)\big[g_{\text{base}}^{(k)}(z_n^{(k)}, t_n) + u_n^{(k)}(\mathcal{Z}_n^c, t_n)\big], \end{aligned} \tag{5}$$

where $g_{\text{base}}^{(k)}$ is the original per-sample guidance and $u_n^{(k)}(\mathcal{Z}_n^c, t_n)$ is our set-level controller. The defining property is that $u_n^{(k)}$ depends on the *full* group state $\mathcal{Z}_n^c$, enabling dense, non-separable coupling at every iteration.

**Controller decomposition.** Shifting from separable to set-coupled dynamics raises two design questions: *what* set-level objective to optimize, and *how* to inject the resulting feedback stably into the diffusion trajectory. We decompose

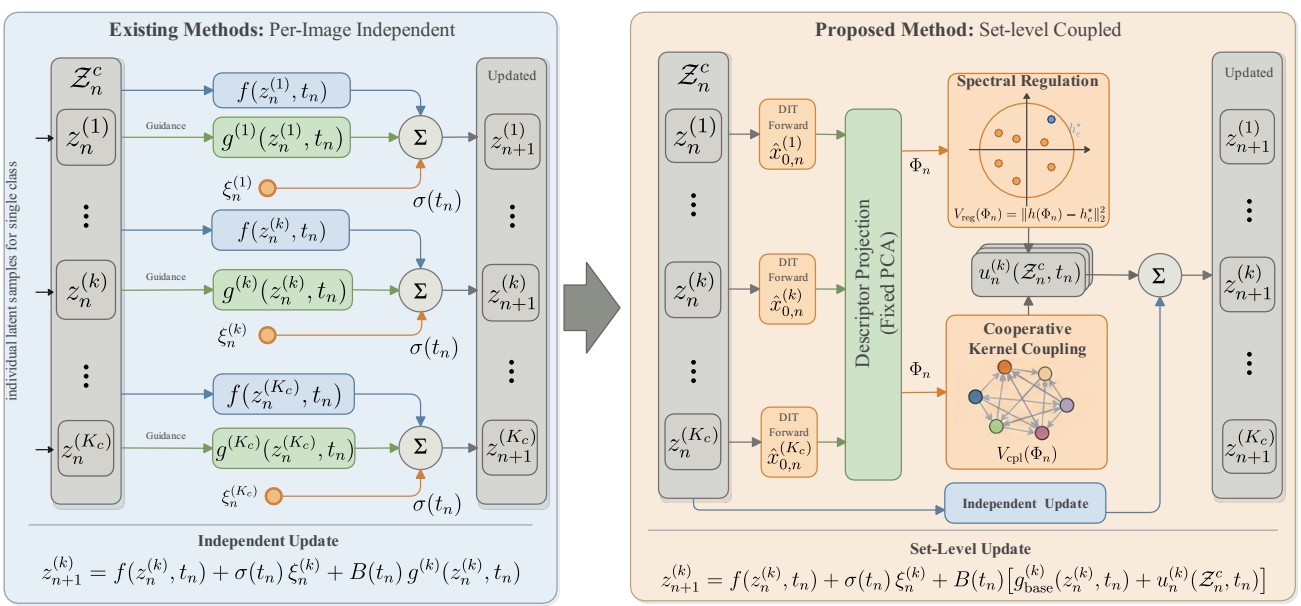

Figure 1. Method overview. Left: per-image (separable) guidance updates each sample independently. Right: SCG computes a *set-level* controller from the full within-class state (via lightweight descriptors) and injects permutation-equivariant feedback to coordinate all IPC samples of a class (IPC-at-once sampling).

the auxiliary controller accordingly:

$$u_n^{(k)}(\mathcal{Z}_n^c, t_n) = \alpha_n\, u_{\text{cpl},n}^{(k)}(\mathcal{Z}_n^c, t_n) + \beta_n\, u_{\text{reg},n}^{(k)}(\mathcal{Z}_n^c, t_n), \quad (6)$$

where the two terms address these questions respectively. Here $\alpha_n = \alpha\,\gamma(t_n)$ and $\beta_n = \beta\,m(t_n)$ are time-varying gains derived from base hyperparameters $\alpha, \beta$ via scheduling functions $\gamma(\cdot)$ (linear ramp) and $m(\cdot)$ (mid-timestep gating); see Section A.5 for details. The *spectral regulation* term $u_{\text{reg}}$ (Section 4.2) defines an explicit distributional alignment objective: it matches the empirical characteristic function (ECF) of the synthetic descriptor set to that of real data, providing analytical, closed-form gradients that encode global set-level statistics rather than pairwise interactions alone. However, this signal is inherently noisy: it operates on indirect $\hat{x}_0$ predictions and passes through an approximate linear lift $\mathcal{P}$, both of which introduce mismatch that can destabilize the joint trajectory. The *cooperative kernel coupling* term $u_{\text{cpl}}$ (Section 4.3) addresses this by regularizing the update geometry: it pulls each sample toward its kernel-weighted neighborhood mean, smoothing the joint dynamics and preventing erratic deviations caused by noisy regulation feedback. We additionally apply root-mean-square (RMS) clipping to bound the auxiliary feedback magnitude (details in Section A.5).

A key design constraint is computational efficiency: differentiating through the diffusion backbone at each step would incur substantial compute and memory overhead. We therefore compute set-level feedback in a low-dimensional descriptor space extracted from the *predicted clean latent* $\hat{x}_{0,n}^{(k)}$—which carries richer semantic content than the

noisy $z_n^{(k)}$—and lift the result back to a latent-shaped update via a linear projection. Specifically, $\hat{x}_{0,n}^{(k)}$ is obtained via the standard DDPM prediction: $\hat{x}_{0,n}^{(k)} = (z_n^{(k)} - \sqrt{1 - \bar{\alpha}_{t_n}}\, \epsilon_\theta(z_n^{(k)}, t_n))/\sqrt{\bar{\alpha}_{t_n}}$. We then project $\hat{x}_{0,n}^{(k)}$ into a low-dimensional descriptor space using a *fixed, class-specific* principal component analysis (PCA) basis $(\mu_c, W_c)$ fitted offline from real latents:

$$\phi_n^{(k)} = \text{normalize}\Big(W_c^\top\big(\text{vec}(\hat{x}_{0,n}^{(k)}) - \mu_c\big)\Big) \in \mathbb{R}^D, \quad (7)$$

where $\text{normalize}(\cdot)$ denotes per-sample $\ell_2$ normalization: for $v \in \mathbb{R}^D$, $\text{normalize}(v) = v/\max(\|v\|_2, \epsilon_{\text{norm}})$ with $\epsilon_{\text{norm}} = 10^{-6}$ (applied independently for each $k$), $\text{vec}(\cdot)$ flattens a latent to $\mathbb{R}^{d_{\text{full}}}$, and $W_c \in \mathbb{R}^{d_{\text{full}} \times D}$ contains the top-$D$ principal components. The basis $(\mu_c, W_c)$ is shared across all $K_c$ samples and kept fixed throughout sampling, ensuring a consistent descriptor space. To inject a descriptor-space direction $q \in \mathbb{R}^D$ back to the latent-shaped update, we use the linear lift $\mathcal{P}(q) = \text{reshape}(W_c q)$.

The joint descriptor $\Phi_n = (\phi_n^{(1)}, \ldots, \phi_n^{(K_c)})$ serves as the observation on which the controller operates.

## 4.2. Spectral Set-Point Regulation

We align the empirical characteristic functions (ECF) of the synthetic and real descriptor sets. The ECF uniquely characterizes a distribution and admits closed-form gradients (Feuerverger & Mureika, 1977). For descriptor set $\Phi_n$,

**Algorithm 1** Set-Coupled Guidance (SCG)

---

**Require:** Base sampler $(f, B, \sigma, g_{\text{base}})$; initial $\{z_0^{(k)}\}_{k=1}^{K_c}$; real set $\mathcal{X}_c$; hyperparameters $\alpha, \beta$

**Ensure:** Distilled latents $\{z_N^{(k)}\}_{k=1}^{K_c}$

1: Fit PCA $(\mu_c, W_c)$; compute target $h_c^\star$ from $\mathcal{X}_c$ (Eq. 9)

2: **for** $n = 0, \dots, N-1$ **do**

3:     Compute time-varying gains (Section A.5):
    $\alpha_n = \alpha \gamma(t_n), \beta_n = \beta m(t_n)$

4:     Predict clean latents $\{\hat{x}_{0,n}^{(k)}\}_{k=1}^{K_c}$

5:     $\phi_n^{(k)} \leftarrow \text{normalize}\big(W_c^\top (\text{vec}(\hat{x}_{0,n}^{(k)}) - \mu_c)\big)$ for all $k$
    (Eq. 7)

6:     $s_n^{(k)} \leftarrow -\nabla_{\phi_n^{(k)}} V_{\text{reg}}(\Phi_n)$ for all $k$      (Eq. 10)

7:     Set bandwidth $h$ and kernel weights (Section A.5):
    $\kappa_{ij} = \kappa(\phi_n^{(i)}, \phi_n^{(j)})$

8:     $r_n^{(k)} \leftarrow \frac{2}{(h+\varepsilon)K_c}\Big(\sum_j \kappa_{kj} \phi_n^{(j)} - \phi_n^{(k)} \sum_j \kappa_{kj}\Big)$ for
    all $k$      (Eq. 13)

9:     $u_n^{(k)} \leftarrow \alpha_n \mathcal{P}(r_n^{(k)}) + \beta_n \mathcal{P}(s_n^{(k)})$ for all $k$   (Eq. 6)

10:     $z_{n+1}^{(k)} \leftarrow f(z_n^{(k)}, t_n) + B(t_n)(g_{\text{base}}^{(k)} + u_n^{(k)}) +$
    $\sigma(t_n) \xi_n^{(k)}$ for all $k$      (Eq. 5)

11: **end for**

---

the ECF is

$$\hat{C}(\omega) = \frac{1}{K_c} \sum_{j=1}^{K_c} e^{i\langle \omega, \phi_n^{(j)} \rangle}. \qquad (8)$$

To form a finite-dimensional statistic, we evaluate $\hat{C}$ at $K_f$ fixed anchor frequencies $\{\omega_\ell\}_{\ell=1}^{K_f}$ sampled once from $\mathcal{N}(0, \Sigma^{-1})$, where $\Sigma$ is the covariance of real-data descriptors, and stack real and imaginary parts:

$$h(\Phi_n) = \big[\text{Re}\,\hat{C}(\omega_\ell), \text{Im}\,\hat{C}(\omega_\ell)\big]_{\ell=1}^{K_f} \in \mathbb{R}^{2K_f}. \qquad (9)$$

The reference $h_c^\star = h(\Phi_c^{\text{real}})$ is the ECF feature vector of real-data descriptors for class $c$, computed once offline and kept fixed. Minimizing $V_{\text{reg}}(\Phi_n) = \|h(\Phi_n) - h_c^\star\|_2^2$ yields a descriptor-space gradient $s_n^{(k)} = -\nabla_{\phi_n^{(k)}} V_{\text{reg}}$ with closed form:

$$s_n^{(k)} = \frac{2}{K_c} \sum_{\ell=1}^{K_f} \big(a_\ell \sin\theta_{k\ell} - b_\ell \cos\theta_{k\ell}\big) \omega_\ell, \qquad (10)$$

where $\theta_{k\ell} = \langle \omega_\ell, \phi_n^{(k)} \rangle$, $\hat{C}^\star$ denotes the ECF of real-data descriptors, $a_\ell = \text{Re}\,\hat{C}(\omega_\ell) - \text{Re}\,\hat{C}^\star(\omega_\ell)$, and $b_\ell = \text{Im}\,\hat{C}(\omega_\ell) - \text{Im}\,\hat{C}^\star(\omega_\ell)$. We lift $s_n^{(k)}$ back to latent space via $\mathcal{P}$:

$$u_{\text{reg},n}^{(k)} \approx \mathcal{P}\big(s_n^{(k)}\big). \qquad (11)$$

### 4.3. Cooperative Kernel Coupling

To implement the stabilizing role of $u_{\text{cpl}}$, we introduce a kernel-based potential that enforces trajectory coherence in

descriptor space.

**Coupling potential.** We define an energy that penalizes low pairwise similarity across the descriptor set $\Phi_n$:

$$V_{\text{cpl}}(\Phi_n) = -\frac{1}{2K_c} \sum_{i=1}^{K_c} \sum_{j=1}^{K_c} \kappa(\phi_n^{(i)}, \phi_n^{(j)}), \qquad (12)$$

where $\kappa(a,b) = \exp(-\|a-b\|^2/(h+\varepsilon))$ is a radial basis function (RBF) kernel with $\varepsilon > 0$ a small constant for numerical stability. We recompute the bandwidth $h$ at each timestep by a detached median heuristic on the current descriptor set (Appendix A.5). Minimizing $V_{\text{cpl}}$ pulls samples toward their kernel-weighted neighborhood means, acting as a stabilizing attraction. Unlike explicit repulsive-kernel interactions that push samples apart, we use coupling to regularize the joint dynamics; distributional alignment is enforced by the spectral regulation term (Section 4.2).

**Coupling direction.** The descriptor-space direction $r_n^{(k)} = -\nabla_{\phi_n^{(k)}} V_{\text{cpl}}(\Phi_n)$ admits a closed-form expression:

$$r_n^{(k)} = \frac{2}{(h+\varepsilon)K_c} \Big(\sum_j \kappa_{kj} \phi_n^{(j)} - \phi_n^{(k)} \sum_j \kappa_{kj}\Big), \quad (13)$$

where $\kappa_{kj} = \kappa(\phi_n^{(k)}, \phi_n^{(j)})$. We lift it back to latent space without backpropagating through the diffusion model:

$$u_{\text{cpl},n}^{(k)} \approx \mathcal{P}\big(r_n^{(k)}\big). \qquad (14)$$

The overall cost is $O(K_c^2 D)$ per class—negligible for typical IPC budgets.

### 4.4. Theoretical Properties

Our analysis targets *measurable set-level objectives* rather than downstream training accuracy, which is generally intractable to certify. All proofs are deferred to Section A.4. For a fixed class $c$, define the composite potential using hyperparameters $\alpha, \beta$:

$$V(\Phi_n) = \alpha\, V_{\text{cpl}}(\Phi_n) + \beta\, V_{\text{reg}}(\Phi_n). \qquad (15)$$

Minimizing $V$ encourages within-class coordination (via $V_{\text{cpl}}$) and distributional fidelity to real data (via $V_{\text{reg}}$). Since $h(\Phi)$ and $V_{\text{cpl}}(\Phi)$ depend only on the unordered set $\{\phi^{(k)}\}$, the resulting update is naturally permutation-equivariant. However, SCG operates on predicted clean latents $\hat{x}_0$ rather than ground-truth images, and the linear lift $\mathcal{P}$ only approximates the true gradient direction. We model these mismatches—together with the base sampler's effect on descriptors—as a bounded perturbation $\Delta_n$ (see Section A.2 for derivation and justification). Under this perturbation model, we establish local Lyapunov descent (Proposition 4.2) and input-to-state stability (ISS) for distributional

*Table 1.* Top-1 Accuracy (%) on ImageNette and ImageWoof with hard-label evaluation. For $CaO_2$, the original paper reports the best of hard/soft-label results; we report hard-label only. **Bold**: SCG improves over base.

| Dataset | Model | IPC | Random | DiT | DiT+SCG | $CaO_2$ | $CaO_2$+SCG | $MGD^3$ | $MGD^3$+SCG | Full |
|---|---|---|---|---|---|---|---|---|---|---|
| ImageNette | ConvNet-6 | 10 | $42.1_{\pm1.3}$ | $55.6_{\pm1.2}$ | $\mathbf{62.2_{\pm0.9}}$ | $58.6_{\pm1.7}$ | $\mathbf{60.8_{\pm0.8}}$ | $58.6_{\pm1.8}$ | $\mathbf{62.0_{\pm1.0}}$ | $92.9_{\pm0.5}$ |
| | | 20 | $49.5_{\pm1.1}$ | $58.2_{\pm1.0}$ | $\mathbf{65.0_{\pm1.5}}$ | $65.2_{\pm1.6}$ | $\mathbf{66.4_{\pm1.7}}$ | $64.4_{\pm1.1}$ | $\mathbf{68.2_{\pm1.7}}$ | |
| | | 50 | $69.1_{\pm0.8}$ | $72.2_{\pm0.9}$ | $\mathbf{79.6_{\pm1.3}}$ | $78.4_{\pm1.2}$ | $\mathbf{78.6_{\pm1.5}}$ | $79.3_{\pm1.2}$ | $\mathbf{81.7_{\pm1.8}}$ | |
| | ResNetAP-10 | 10 | $47.3_{\pm1.5}$ | $60.6_{\pm1.4}$ | $\mathbf{65.2_{\pm1.0}}$ | $62.0_{\pm1.6}$ | $61.4_{\pm1.2}$ | $65.1_{\pm0.6}$ | $\mathbf{67.2_{\pm0.6}}$ | $93.7_{\pm0.4}$ |
| | | 20 | $55.4_{\pm1.2}$ | $62.6_{\pm1.1}$ | $\mathbf{71.2_{\pm0.8}}$ | $70.2_{\pm1.3}$ | $\mathbf{70.8_{\pm0.7}}$ | $70.7_{\pm0.9}$ | $\mathbf{75.4_{\pm1.9}}$ | |
| | | 50 | $72.3_{\pm0.9}$ | $71.8_{\pm0.8}$ | $\mathbf{80.8_{\pm1.7}}$ | $75.4_{\pm1.3}$ | $\mathbf{77.8_{\pm1.9}}$ | $78.1_{\pm1.3}$ | $\mathbf{82.8_{\pm1.5}}$ | |
| | ResNet-18 | 10 | $45.3_{\pm1.4}$ | $58.2_{\pm1.3}$ | $\mathbf{65.8_{\pm1.3}}$ | $61.5_{\pm1.2}$ | $\mathbf{62.8_{\pm1.1}}$ | $61.5_{\pm0.7}$ | $\mathbf{64.1_{\pm1.6}}$ | $92.4_{\pm0.6}$ |
| | | 20 | $53.6_{\pm1.0}$ | $62.0_{\pm1.0}$ | $\mathbf{69.4_{\pm1.4}}$ | $70.0_{\pm1.1}$ | $\mathbf{70.6_{\pm1.6}}$ | $69.1_{\pm1.4}$ | $\mathbf{73.0_{\pm1.5}}$ | |
| | | 50 | $70.4_{\pm0.7}$ | $71.6_{\pm0.7}$ | $\mathbf{79.6_{\pm1.4}}$ | $73.8_{\pm1.0}$ | $\mathbf{76.6_{\pm1.2}}$ | $78.1_{\pm0.4}$ | $\mathbf{82.6_{\pm1.9}}$ | |
| ImageWoof | ConvNet-6 | 10 | $26.7_{\pm1.2}$ | $31.0_{\pm1.1}$ | $\mathbf{33.2_{\pm0.5}}$ | $34.4_{\pm0.8}$ | $\mathbf{35.2_{\pm0.4}}$ | $33.3_{\pm0.8}$ | $\mathbf{34.0_{\pm1.0}}$ | $83.4_{\pm0.7}$ |
| | | 20 | $31.4_{\pm1.1}$ | $33.2_{\pm0.9}$ | $\mathbf{38.8_{\pm1.3}}$ | $38.6_{\pm1.1}$ | $\mathbf{41.0_{\pm1.1}}$ | $37.3_{\pm1.1}$ | $\mathbf{39.2_{\pm1.7}}$ | |
| | | 50 | $40.1_{\pm1.3}$ | $45.6_{\pm1.2}$ | $\mathbf{52.4_{\pm1.2}}$ | $49.4_{\pm1.7}$ | $\mathbf{52.0_{\pm1.4}}$ | $51.1_{\pm1.1}$ | $\mathbf{52.6_{\pm1.2}}$ | |
| | ResNetAP-10 | 10 | $30.1_{\pm1.4}$ | $31.4_{\pm1.0}$ | $\mathbf{36.8_{\pm1.4}}$ | $35.4_{\pm1.3}$ | $\mathbf{36.8_{\pm1.6}}$ | $37.3_{\pm1.8}$ | $\mathbf{39.2_{\pm1.1}}$ | $87.7_{\pm0.5}$ |
| | | 20 | $35.6_{\pm1.3}$ | $40.2_{\pm1.3}$ | $\mathbf{42.8_{\pm1.5}}$ | $42.6_{\pm0.4}$ | $\mathbf{43.0_{\pm1.7}}$ | $45.9_{\pm1.3}$ | $45.4_{\pm1.4}$ | |
| | | 50 | $50.3_{\pm1.0}$ | $51.0_{\pm0.9}$ | $\mathbf{55.4_{\pm1.3}}$ | $54.8_{\pm1.4}$ | $\mathbf{55.2_{\pm1.5}}$ | $56.7_{\pm1.5}$ | $\mathbf{59.6_{\pm1.3}}$ | |
| | ResNet-18 | 10 | $26.8_{\pm1.3}$ | $29.0_{\pm1.2}$ | $\mathbf{36.6_{\pm1.1}}$ | $35.0_{\pm0.9}$ | $\mathbf{38.2_{\pm1.3}}$ | $36.8_{\pm0.9}$ | $\mathbf{38.2_{\pm1.4}}$ | $82.8_{\pm0.8}$ |
| | | 20 | $32.3_{\pm1.2}$ | $35.2_{\pm1.1}$ | $\mathbf{41.0_{\pm1.0}}$ | $39.4_{\pm1.1}$ | $\mathbf{42.6_{\pm1.2}}$ | $42.5_{\pm1.1}$ | $\mathbf{43.6_{\pm0.8}}$ | |
| | | 50 | $51.0_{\pm0.9}$ | $50.2_{\pm0.8}$ | $\mathbf{60.0_{\pm1.6}}$ | $51.4_{\pm1.2}$ | $\mathbf{53.0_{\pm1.8}}$ | $58.3_{\pm0.7}$ | $58.0_{\pm0.6}$ | |

tracking (Proposition 4.3) using standard nonlinear systems techniques (Khalil, 2002; Sontag, 2008). Finally, we show that optimizing non-additive set objectives (i.e., objectives not decomposable as a sum over samples) inherently requires coupled gradients, formalizing the limitation of separable dynamics.

**Lemma 4.1** (Separability implies additivity). *Let* $\Phi = (\phi^{(1)}, \ldots, \phi^{(K_c)})$ *and consider a separable vector field* $G(\Phi) = (G^{(1)}, \ldots, G^{(K_c)})$ *where* $G^{(k)}$ *depends only on* $\phi^{(k)}$. *If* $G(\Phi) = -\nabla_\Phi U(\Phi)$ *for some scalar potential* $U$, *then* $U$ *must be additive:* $U(\Phi) = \sum_{k=1}^{K_c} \psi_k(\phi^{(k)}) + \text{const.}$

This result implies that strictly per-sample gradient descent cannot optimize objectives like $V_{\text{reg}}$ (which involves ECF averaging) or $V_{\text{cpl}}$ (pairwise kernels), as their gradients are inherently non-separable.

**Proposition 4.2** (Lyapunov decrease). *Because descriptors are* $\ell_2$*-normalized (Eq. 7), the descriptor configuration lies on the product manifold* $\mathcal{M} = (\mathbb{S}^{D-1})^{K_c}$. *With time-varying gains* $\alpha_n, \beta_n$ *and RMS clipping (Section A.5), the effective perturbation satisfies* $\|\Delta_n\| \leq \bar{\Delta}$ *for some* $\bar{\Delta} > 0$. *Under standard smoothness conditions (Section A.4), after* $N$ *update steps the time-averaged squared gradient norm*

*satisfies:*

$$\frac{1}{N} \sum_{n=0}^{N-1} \|\nabla_{\mathcal{M}} V(\Phi_n)\|^2 \leq \frac{C_0}{N} + C_1 \bar{\Delta}^2, \quad (16)$$

*for constants* $C_0, C_1 > 0$ *depending on the initial potential gap and smoothness.*

In other words, the controller makes steady progress toward minimizing the set-level objective $V$: the first term vanishes as $N \to \infty$, while the second term bounds the residual error due to operating on noisy predicted latents.

**Proposition 4.3** (ISS for distributional tracking). *Let* $e_n = h(\Phi_n) - h_c^\star$ *denote the ECF tracking error between synthetic and real descriptors. Within the regulation-active window (where* $m(t_n) = 1$ *and thus* $\beta_n = \beta$*), if the ECF Jacobian is locally well-conditioned (*$J_h J_h^\top \succeq \mu I$ *for some* $\mu > 0$*) and the coupling gain is annealed (*$\alpha_n \to 0$*), then the error contracts geometrically up to perturbations:*

$$\|e_{n+1}\| \leq (1 - \mu\eta\beta)\|e_n\| + \eta\alpha_n C_{cpl} + O(\bar{\Delta}), \quad (17)$$

*where* $\eta$ *is the (effective) step size, corresponding to the input gain* $B(t_n)$ *in (4) (e.g., for* $CaO_2$*,* $\eta = B(t_n)$*), and* $C_{cpl}$ *bounds the coupling contribution.*

This guarantees that the synthetic set's distributional statistics track the real-data reference: the contraction factor

*Table 2.* Top-1 Accuracy (%) on ImageNet-100 with hard-label evaluation. **Bold**: SCG improves over base.

| Method | IPC=10 | | | IPC=20 | | |
|---|---|---|---|---|---|---|
| | ConvNet-6 | ResNetAP-10 | ResNet-18 | ConvNet-6 | ResNetAP-10 | ResNet-18 |
| Random | $16.8_{\pm 0.4}$ | $19.4_{\pm 0.3}$ | $17.2_{\pm 0.6}$ | $25.1_{\pm 0.3}$ | $26.4_{\pm 0.6}$ | $25.2_{\pm 0.4}$ |
| $CaO_2$ | $24.3_{\pm 0.7}$ | $25.8_{\pm 0.9}$ | $25.1_{\pm 0.6}$ | $30.2_{\pm 1.3}$ | $31.3_{\pm 1.1}$ | $31.9_{\pm 1.2}$ |
| $CaO_2$+SCG | $\mathbf{24.9_{\pm 0.9}}$ | $\mathbf{26.4_{\pm 0.7}}$ | $25.1_{\pm 1.1}$ | $\mathbf{32.0_{\pm 1.1}}$ | $\mathbf{32.5_{\pm 1.3}}$ | $\mathbf{32.3_{\pm 0.8}}$ |
| $MGD^3$ | $23.2_{\pm 1.2}$ | $24.9_{\pm 0.9}$ | $23.4_{\pm 0.8}$ | $30.2_{\pm 0.7}$ | $33.4_{\pm 1.2}$ | $33.3_{\pm 0.5}$ |
| $MGD^3$+SCG | $\mathbf{25.0_{\pm 1.1}}$ | $\mathbf{27.1_{\pm 0.6}}$ | $\mathbf{24.6_{\pm 0.7}}$ | $\mathbf{31.8_{\pm 0.6}}$ | $\mathbf{35.3_{\pm 0.9}}$ | $\mathbf{35.0_{\pm 1.3}}$ |
| Full Dataset | $79.6_{\pm 0.5}$ | $80.6_{\pm 0.3}$ | $81.5_{\pm 0.8}$ | $79.6_{\pm 0.5}$ | $80.6_{\pm 0.3}$ | $81.5_{\pm 0.8}$ |

*Table 3.* Top-1 Accuracy (%) on ImageNet-1K with hard-label evaluation (IPC=10). Results are from a single run. **Bold**: SCG improves over base.

| Method | ConvNet-6 | ResNetAP-10 | ResNet-18 |
|---|---|---|---|
| DiT | 10.3 | 14.1 | 14.7 |
| DiT+SCG | **12.1** | **16.4** | **17.2** |
| $MGD^3$ | 13.1 | 20.7 | 21.2 |
| $MGD^3$+SCG | **14.2** | **21.3** | **22**.9 |

$(1 - \mu\eta\beta) < 1$ drives $e_n$ toward zero, while the latter two terms bound the residual from coupling and approximation mismatch. The well-conditioning assumption ($K_c D \geq 2K_f$ with non-degenerate anchors) is mild when anchor frequencies are randomly sampled (Section A.3).

**Proposition 4.4** (Anti-collapse guarantee)**.** *Suppose the real class distribution is non-degenerate, i.e., there exists an anchor $\omega_\ell$ with $|\hat{C}^\star(\omega_\ell)| \leq 1 - \rho$ for some $\rho > 0$. If the synthetic descriptor set collapses to a single point ($\phi_n^{(1)} = \cdots = \phi_n^{(K_c)}$), then $|\hat{C}(\omega)| = 1$ for all $\omega$, and the regulation penalty is bounded below:*

$$V_{reg}(\Phi_n) \geq \rho^2. \tag{18}$$

In other words, complete collapse incurs a strictly positive cost whenever real data has any spread—providing an implicit diversity incentive. Since any non-point-mass distribution satisfies $|\hat{C}^\star(\omega)| < 1$ for some $\omega \neq 0$, random anchor sampling ensures the premise holds with high probability as $K_f$ grows.

## 5. Experiments

We evaluate SCG as a plug-and-play module integrated into three representative diffusion-based DD pipelines: a DiT baseline with vanilla sampling, $MGD^3$ (Santiago et al., 2025), and $CaO_2$ (Wang et al., 2025a); results on IGD (Chen et al., 2025) are provided in Section B.7.

### 5.1. Experimental Settings

**Datasets.** We conduct experiments on ImageNet-1K (Deng et al., 2009) and its subsets of varying difficulty: ImageNette (Howard, 2019a), a 10-class subset with visually distinct categories; ImageWoof (Howard, 2019b), a challenging 10-class subset of visually similar dog breeds; and ImageNet-100 (Tian et al., 2020), a 100-class subset. All images are at $256 \times 256$ resolution. For ImageNette and ImageWoof, we adopt IPC$\in \{10, 20, 50\}$; for ImageNet-100, IPC$\in \{10, 20\}$; and for ImageNet-1K, IPC$= 10$.

**Evaluation protocol.** Following prior work (Gu et al., 2024; Santiago et al., 2025), we use the hard-label protocol: classifiers are trained from scratch on distilled data with ground-truth labels and evaluated on the original test set. Note that $CaO_2$ reports the best of hard/soft-label results in its original paper (Wang et al., 2025a); we report hard-label only for all methods. For ImageNette/ImageWoof, we train for 2000 epochs (IPC=10) or 1500 epochs (IPC$\geq$20) using SGD with learning rate 0.01 and decay at 2/3 and 5/6 of training. For ImageNet-100, we follow the same protocol with IPC$\in \{10, 20\}$. Models are evaluated on three architectures following prior work (Santiago et al., 2025; Gu et al., 2024): ConvNet-6, ResNetAP-10, and ResNet-18. Experiments on ImageNette, ImageWoof, and ImageNet-100 are repeated three times and we report mean $\pm$ std. For ImageNet-1K, we report results from a single run due to the computational cost.

**Implementation details.** We integrate SCG into the official codebases of $MGD^3$ and $CaO_2$. For the DiT baseline, we generate distilled data via vanilla DDPM sampling without any additional guidance or post-hoc refinement. The pre-trained DiT-XL/2-256 (Peebles & Xie, 2023) serves as the diffusion backbone. For the descriptor space, we use a class-wise PCA projection with $D = 64$ dimensions, fitting $(\mu_c, W_c)$ once from real VAE latents per class and reusing it during sampling. Anchor frequencies $K_f = 32$ are sampled once from $\mathcal{N}(0, \Sigma^{-1})$ and fixed throughout, where $\Sigma$ is the covariance of real-data descriptors; see Appendix A.3 for motivation and theoretical interpretation. Default hyperpa-

*Table 4.* Computational overhead on ImageNette (IPC=10, per class). Comparing batched baselines to +SCG isolates controller overhead (< 2%).

| Method | Time (s) | GFLOPs | TFLOPs/s |
|---|---|---|---|
| $MGD^3$ (seq.) | 10.88 | 981,038 | 90.2 |
| $MGD^3$ (batched) | 8.20 | 981,038 | 119.6 |
| $MGD^3$+SCG (batched) | 8.35 | 981,038 | 117.5 |
| $CaO_2$ (seq.) | 56.79 | 16,735 | 0.295 |
| $CaO_2$ (batched) | 16.11 | 16,735 | 1.039 |
| $CaO_2$+SCG (batched) | 16.42 | 16,735 | 1.019 |

*Table 5.* IPC=50 cross-architecture evaluation on larger backbones. R50/R101 denote ResNet-50/101, VS/VB/VL denote ViT-S/B/L, and SwB denotes Swin-B.

| Dataset | Method | R50 | R101 | VS | VB | VL | SwB | Mean |
|---|---|---|---|---|---|---|---|---|
| | $MGD^3$ | 73.8 | 70.4 | 59.2 | 62.0 | 64.2 | 69.4 | 66.5 |
| ImageNette | +SCG | **75.0** | **74.0** | **59.4** | **65.8** | **67.8** | **70.2** | **68.7** |
| | Gain | +1.2 | +3.6 | +0.2 | +3.8 | +3.6 | +0.8 | +2.2 |
| | $MGD^3$ | 43.6 | 40.0 | 34.8 | 39.2 | 38.2 | 38.6 | 39.1 |
| ImageWoof | +SCG | **46.0** | **41.2** | **37.2** | **40.2** | **39.2** | **40.2** | **40.7** |
| | Gain | +2.4 | +1.2 | +2.4 | +1.0 | +1.0 | +1.6 | +1.6 |

rameters are $\alpha = 1.0$, $\beta = 0.15$, and RMS clipping ratio $r_{max} = 2.0$ (see Appendix A.5). All experiments are conducted on a single NVIDIA RTX 5090 GPU with PyTorch 2.9.1.

## 5.2. Main Results

Table 1 summarizes Top-1 accuracy on ImageNette and ImageWoof under hard-label evaluation. SCG consistently boosts vanilla DiT sampling and yields additional gains on top of $CaO_2$ and $MGD^3$ in most settings. The gains are larger on the harder ImageWoof benchmark and at small-to-medium IPC, where per-sample generation is more prone to redundancy and the evaluator is not yet saturated. When stacked on $CaO_2$ and $MGD^3$, SCG is often complementary; we analyze the component contributions in Section 5.4.

**Larger cross-architecture evaluators.** Table 5 stress-tests IPC=50 distilled sets on substantially larger backbones than the standard DD evaluators, covering ResNet (He et al., 2016), Vision Transformer (ViT) (Dosovitskiy et al., 2021), and Swin Transformer (Liu et al., 2021) families. Across ResNet-50/101, ViT-S/B/L, and Swin-Base, SCG improves every $MGD^3$ evaluation: the six-architecture mean increases from 66.5% to 68.7% on ImageNette and from 39.1% to 40.7% on ImageWoof. These single-seed evaluations indicate that the set-coupled controller transfers beyond ConvNet/ResNet-18-scale architectures.

**ImageNet-100.** Table 2 reports results on the larger-scale ImageNet-100 benchmark. SCG further improves $MGD^3$ and improves or matches $CaO_2$, with gains up to 2.2% on

ResNetAP-10 at IPC=10. **ImageNet-1K.** Table 3 reports results on the full ImageNet-1K benchmark at IPC=10 with hard-label evaluation. We report single-run results. SCG improves both DiT and $MGD^3$ baselines across all architectures.

## 5.3. Computational Overhead

Table 4 reports per-class wall-clock time and throughput on ImageNette (IPC=10). $MGD^3$ uses 50-step sampling; $CaO_2$ reports stage-2 only with latent_steps=100. GFLOPs measures diffusion-backbone compute and is identical across batching modes. To isolate the overhead introduced by the SCG controller, we compare three configurations: sequential baseline (one sample per forward pass), batched baseline (IPC samples per forward pass), and batched+SCG. The transition from sequential to batched execution yields substantial speedups by improving hardware utilization (e.g., $CaO_2$: 56.79s → 16.11s). Crucially, adding SCG on top of the batched baseline incurs negligible additional cost: +0.15s for $MGD^3$ and +0.31s for $CaO_2$, corresponding to < 2% overhead in both cases. This confirms that the set-level controller is lightweight relative to backbone computation.

## 5.4. Ablation Study

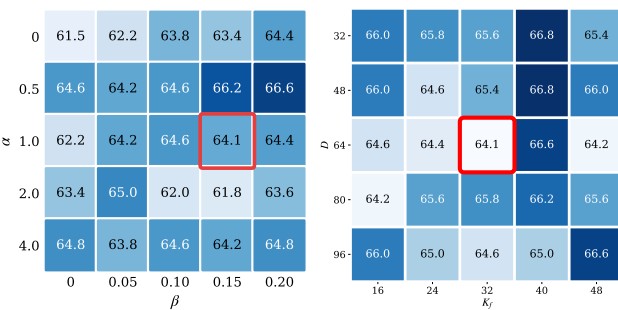

(a) Controller weights $(\alpha, \beta)$.  (b) Descriptor/anchor settings $(D, K_f)$.

*Figure 2.* Hyperparameter sensitivity on ImageNette (IPC=10) with ResNet-18 evaluation; colors indicate top-1 accuracy (%).

Table 6 ablates the two SCG components on $MGD^3$+SCG using ImageNette at IPC=10. Both the kernel coupling ($\alpha$) and spectral regulation ($\beta$) contribute to the overall gain, with their combination yielding the best performance.

*Table 6.* Ablation on SCG components (ImageNette, IPC=10).

| $\alpha$ | $\beta$ | ConvNet-6 | ResNetAP-10 | ResNet-18 |
|---|---|---|---|---|
| – | – | $58.6_{\pm 1.8}$ | $65.1_{\pm 0.6}$ | $61.5_{\pm 0.7}$ |
| ✓ | – | $60.0_{\pm 1.0}$ | $64.2_{\pm 0.8}$ | $62.2_{\pm 0.9}$ |
| – | ✓ | $61.2_{\pm 0.9}$ | $65.8_{\pm 0.7}$ | $63.4_{\pm 1.1}$ |
| ✓ | ✓ | $\mathbf{62.0_{\pm 1.0}}$ | $\mathbf{67.2_{\pm 0.6}}$ | $\mathbf{64.1_{\pm 1.6}}$ |

Spectral regulation ($\beta$) provides the dominant gains when

used alone, while kernel coupling ($\alpha$) is most effective when combined with regulation, yielding the best performance across all three architectures. Set-level diagnostics (Figure 3 in Section B.2) further confirm monotonic CF-distance reduction and stable diversity throughout sampling, empirically validating the Lyapunov descent and anti-collapse properties established in Section 4.4.

**Hyperparameter Sensitivity.** We sweep the controller weights $(\alpha, \beta)$ on ImageNette (IPC=10) with ResNet-18 evaluation. All main results use the fixed default ($\alpha = 1.0, \beta = 0.15$) without per-setting tuning; in Figure 2a, the red-boxed cell marks this default and the heatmap shows a broad plateau around it, suggesting SCG is not overly sensitive to these weights in this setting. We also sweep the descriptor dimension $D$ and the number of anchor frequencies $K_f$ used in the empirical characteristic-function (ECF) features (keeping $(\alpha, \beta)$ fixed) and observe a non-monotonic interaction; in this sweep, the range spans $64.1\%$ to $66.8\%$ top-1, indicating additional headroom from tuning $D$ and $K_f$ beyond our compute-balanced default ($D=64$, $K_f=32$).

## 6. Conclusion

We introduced *Set-Coupled Guidance* (SCG), a plug-and-play controller that shifts diffusion-based dataset distillation from independent per-image sampling to *IPC-at-once* set-level generation. By casting existing methods into a unified state-space form, we highlight separability as a key limitation for optimizing non-additive set objectives and address it via permutation-equivariant feedback on lightweight descriptors. Theoretically, SCG comes with Lyapunov descent and ISS guarantees; empirically, it yields consistent gains across ImageNette, ImageWoof, ImageNet-100 and ImageNet-1K and three base methods, with negligible controller cost (e.g., $< 2\%$ overhead on ImageNette). Overall, SCG offers a simple and principled way to inject set-level objectives into diffusion samplers, helping scale controllable synthetic-set generation without altering the backbone model. Set-level diagnostics further suggest monotonic CF-distance reduction with stable diversity, helping mitigate redundancy during generation, especially at small-to-medium IPC.

## Acknowledgements

The authors thank the anonymous reviewers for their valuable feedback on earlier versions of this manuscript.

## Impact Statement

This paper aims to advance dataset distillation by reducing the storage and compute needed for visual learning experiments. As with other distillation and generative-data methods, synthetic sets may inherit biases or privacy risks from source data and pretrained generators, so sensitive deployments should include appropriate data-governance checks.

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

# A. Additional Details and Proofs

## A.1. Instantiation of the Unified Framework

We show how existing diffusion-based dataset distillation methods that provide a per-sample update direction instantiate the general state-space form in (4):

$$z_{n+1}^{(k)} = f(z_n^{(k)}, t_n) + B(t_n)\, g^{(k)}(z_n^{(k)}, t_n) + \sigma(t_n)\, \xi_n^{(k)}.$$

**MGD$^3$ (Mode-Guided Sampling).** MGD$^3$ (Santiago et al., 2025) modifies the DDPM reverse process by injecting a mode guidance term that steers each sample toward a pre-discovered cluster centroid. At each reverse step $t$, the method first computes the predicted clean latent $\hat{x}_{0,t}$ from the current noisy state $z_t$ via the standard DDPM formula, then constructs a guidance direction $g_t = m_i - \hat{x}_{0,t}$ pointing toward the assigned mode $m_i$. In the implementation used in our experiments, this guidance is added to the reverse mean, yielding the update

$$z_{t-1} = \mu_\theta(z_t, t, c) + \lambda\, \sigma_t\, g_t + \sigma_t\, \xi_t,$$

where $\lambda$ controls guidance strength and $\sigma_t$ is the standard DDPM noise scale.

Mapping to (4): $f(z_t, t) + \sigma(t)\xi$ is the DDPM reverse step (3), with $f(z_t, t) = \frac{1}{\sqrt{\alpha_t}}\big(z_t - \frac{\beta_t}{\sqrt{1-\bar{\alpha}_t}}\epsilon_\theta(z_t, t)\big)$ and $\sigma(t) = \sigma_t$. Substituting $\mu_\theta(z_t, t, c) = \frac{1}{\sqrt{\alpha_t}}\big(z_t - \frac{\beta_t}{\sqrt{1-\bar{\alpha}_t}}\epsilon_\theta(z_t, t, c)\big)$ into the update above shows that the guidance enters linearly as $+B(t)\, g^{(k)}$ with input gain $B(t) = \lambda\sigma_t$. The per-sample guidance $g^{(k)}$ corresponds to the mode direction $m_i - \hat{x}_{0,t}$, and $\sigma(t_n) = \sigma_t > 0$ is the standard DDPM stochastic noise coefficient.

**IGD (Influence-Guided Diffusion).** IGD (Chen et al., 2025) computes a per-sample guidance direction based on the influence of each generated sample on downstream classifier training. At each reverse step, IGD estimates how the current predicted clean image $\hat{x}_0$ would affect the validation loss if added to the training set, and uses this gradient signal to steer generation toward more informative samples. Additionally, IGD introduces a deviation (DEV) term that conditions each sample on previously generated ones to encourage diversity.

Mapping to (4): similar to MGD$^3$, $f(z, t) + \sigma(t)\xi$ is the DDPM reverse step (3). The guidance $g^{(k)}$ is the influence-based gradient direction (and optionally the DEV term), scaled by a time-dependent coefficient absorbed into $B(t_n)$. The diffusion coefficient $\sigma(t_n) > 0$ follows the standard DDPM schedule. Note that the DEV term introduces a sequential dependency on previously generated samples, forming a causal (lower-triangular) coupling rather than the dense coupling provided by our SCG controller.

**CaO$_2$ (Latent Refinement).** Unlike guided sampling methods that modify the reverse diffusion trajectory, CaO$_2$ (Wang et al., 2025a) operates on *static* latent codes. Starting from latents obtained via standard diffusion sampling, CaO$_2$ iteratively refines them by minimizing the diffusion loss with respect to the latent input:

$$\min_z\ \mathbb{E}_{t,\epsilon}\big[\|\epsilon_\theta(z_t, \hat{c}, t) - \epsilon\|_2^2\big], \quad \text{where } z_t = \sqrt{\bar{\alpha}_t}\, z + \sqrt{1-\bar{\alpha}_t}\, \epsilon.$$

This optimization is performed via gradient descent on $z$, with $t$ randomly sampled at each iteration to form a stochastic update direction.

Mapping to (4): since the latent is directly updated rather than evolved through reverse diffusion, $f(z, t) = z$ and $\sigma(t) = 0$. The input gain $B(t_n)$ corresponds to the optimizer step size (learning rate) $\eta$. The per-sample guidance $g^{(k)} = -\nabla_z \mathbb{E}_{t,\epsilon}\big[\|\epsilon_\theta(z_t, \hat{c}, t) - \epsilon\|^2\big]$ is the negative gradient of the diffusion loss. Crucially, $\sigma(t_n) = 0$—there is no injected stochasticity in the update rule itself (though $t$ is randomly sampled to compute the gradient direction).

**Where Does $\hat{x}_0$ Come From in CaO$_2$+SCG?** In CaO$_2$ stage-2, the optimized variable $z$ plays the role of the clean latent ($x_0$), and the randomly sampled timestep $t$ is used only to construct $z_t$ (via forward noising) and the diffusion-loss gradient. Therefore, in our CaO$_2$+SCG implementation, we take $\hat{x}_0 \triangleq z$ and compute the set-level controller directly on the current latent set $\{z^{(k)}\}_{k=1}^{K_c}$, without introducing an additional "predict $\hat{x}_0$ from $z_t$" step. The sampled timesteps $t$ are used only to form the base diffusion-loss gradient and to compute a scalar timestep weight for downweighting the auxiliary update near clean timesteps.

**Summary.** Despite their algorithmic differences, all three methods fit the unified form (4) with method-specific instantiations of $(f, B, g, \sigma)$. This abstraction allows our SCG controller to be added as a plug-in term $u_n^{(k)}(\mathcal{Z}_n^c, t_n)$ without modifying the base method's update logic, enabling set-level coordination on top of any compatible backbone.

### A.2. Derivation of Error Bounds

The bounded disturbance assumption in Section 4.4 holds under mild regularity conditions on the diffusion backbone and the descriptor map. Specifically, the mismatch $\Delta_n$ aggregates three sources of error:

1. **Base/backbone-induced drift and observation mismatch:** The descriptors are computed from $\hat{x}_0$ predicted on the current latent, which itself is advanced by the base sampler (drift $f$, per-sample guidance $g_{\text{base}}$, and diffusion noise $\sigma\xi$). These base dynamics, together with noise prediction error in $\hat{x}_0$, induce a descriptor-space deviation from an idealized gradient step. For DDPM-style reverse steps, the magnitude of these effects scales with the noise level (e.g., $O(\sigma_t) = O(\sqrt{1 - \bar{\alpha}_t})$) after the input gain and clipping/gating.

2. **Descriptor extraction error:** The PCA projection $\phi(\cdot)$ in (7) is a linear map composed with normalization. Since images lie in a compact domain and the PCA basis $(W_c, \mu_c)$ is fixed, the descriptor map is Lipschitz-continuous.

3. **Lift mismatch:** The linear lift $\mathcal{P}(q) = \text{reshape}(W_c q)$ is an approximate inverse of the projection. The mismatch $\epsilon_{\mathcal{P}}$ captures the component of the true gradient that lies outside the column space of $W_c$ and generally need not vanish.

Under these conditions, there exist constants $c_1, c_2 > 0$ such that $\|\Delta_n\| \le c_1 \epsilon(t_n) + c_2 \epsilon_{\mathcal{P}}(t_n)$, where $\epsilon(t_n)$ captures the aggregate base/backbone-induced descriptor drift and observation/descriptor mismatch. For DDPM-style reverse steps it is natural to relate this term to the noise scale $\sigma(t_n)$ (e.g., $\epsilon(t_n) = O(\sigma(t_n)) = O(\sqrt{1 - \bar{\alpha}_{t_n}})$ up to model error and discretization); we do not assume $\epsilon(t_n)$ or $\epsilon_{\mathcal{P}}(t_n)$ vanishes, only that they remain bounded. The gain scheduling and RMS clipping described in Section A.5 further ensure that the effective disturbance remains bounded throughout the trajectory.

### A.3. Why Sample Anchor Frequencies from $\mathcal{N}(0, \Sigma^{-1})$?

Let $P$ and $Q$ denote the real and synthetic descriptor distributions in $\mathbb{R}^D$, with characteristic functions $\varphi_P(\omega) = \mathbb{E}_{x \sim P}[e^{i\langle\omega, x\rangle}]$ and $\varphi_Q(\omega) = \mathbb{E}_{y \sim Q}[e^{i\langle\omega, y\rangle}]$. This section provides theoretical justification for the anchor sampling strategy used in spectral regulation. The connection to random Fourier features (Rahimi & Recht, 2007) and energy statistics (Székely & Rizzo, 2013) motivates our design choices.

A. WEIGHTED CF DISCREPANCY EQUALS A GAUSSIAN-KERNEL MMD

The maximum mean discrepancy (MMD) (Gretton et al., 2012) is a kernel-based metric that measures the distance between two probability distributions. We show that sampling anchors from $\mathcal{N}(0, \Sigma^{-1})$ yields an ECF objective equivalent to a Gaussian-kernel MMD.

Consider the weighted discrepancy

$$\mathcal{D}_p^2(P, Q) \triangleq \int_{\mathbb{R}^D} |\varphi_P(\omega) - \varphi_Q(\omega)|^2 p(\omega)\, d\omega, \qquad p(\omega) = \mathcal{N}(0, \Sigma^{-1}). \tag{19}$$

Expanding the squared magnitude and exchanging integrals with expectations yields

$$\mathcal{D}_p^2(P, Q) = \mathbb{E}_{x,x' \sim P}\left[\int e^{i\langle\omega, x-x'\rangle} p(\omega)\, d\omega\right] + \mathbb{E}_{y,y' \sim Q}\left[\int e^{i\langle\omega, y-y'\rangle} p(\omega)\, d\omega\right]$$
$$- 2\,\mathbb{E}_{x \sim P, y \sim Q}\left[\int e^{i\langle\omega, x-y\rangle} p(\omega)\, d\omega\right].$$

For $\omega \sim \mathcal{N}(0, \Sigma^{-1})$ and any $\delta \in \mathbb{R}^D$, the Gaussian integral is the characteristic function of a Gaussian:

$$\int e^{i\langle\omega, \delta\rangle} p(\omega)\, d\omega = \mathbb{E}_\omega[e^{i\langle\omega, \delta\rangle}] = \exp\left(-\tfrac{1}{2}\delta^\top \Sigma^{-1} \delta\right). \tag{20}$$

Define the shift-invariant Gaussian kernel with precision matrix $\Lambda = \Sigma^{-1}$ (distinct from the RBF kernel $\kappa$ used in cooperative coupling):

$$k_\Lambda(u, v) \triangleq \exp\left(-\tfrac{1}{2}(u - v)^\top \Lambda (u - v)\right). \tag{21}$$

Substituting (20) into the expansion shows

$$\mathcal{D}_p^2(P, Q) = \mathbb{E}_{x,x' \sim P}[k_\Lambda(x, x')] + \mathbb{E}_{y,y' \sim Q}[k_\Lambda(y, y')] - 2\, \mathbb{E}_{x \sim P, y \sim Q}[k_\Lambda(x, y)] = \mathrm{MMD}_{k_\Lambda}^2(P, Q). \tag{22}$$

Hence, sampling anchor frequencies from $\mathcal{N}(0, \Sigma^{-1})$ corresponds to approximating a Gaussian-kernel MMD in descriptor space, with anisotropy adapted to the real-data covariance. This connection to random Fourier features (Rahimi & Recht, 2007) provides a principled basis for finite-anchor approximation.

## C. Monte Carlo Approximation and Finite-Sample Bounds

Let $\omega_\ell \overset{i.i.d.}{\sim} p(\omega)$ for $\ell = 1, \ldots, K_f$. The empirical objective $\hat{\mathcal{D}}_{K_f}^2 = \frac{1}{K_f} \sum_{\ell=1}^{K_f} |\hat{C}(\omega_\ell) - \hat{C}^\star(\omega_\ell)|^2$ is a Monte Carlo approximation of (19). By standard concentration arguments for bounded random variables (since $|\hat{C}(\omega) - \hat{C}^\star(\omega)| \leq 2$), we have with probability at least $1 - \delta$:

$$\left|\hat{\mathcal{D}}_{K_f}^2 - \mathcal{D}_p^2(P, Q)\right| \leq 4\sqrt{\frac{\log(2/\delta)}{2K_f}}. \tag{23}$$

For $K_f = 32$ (our default), this yields $\approx 0.96$ approximation error at 95% confidence, noting that such worst-case bounds can be loose and mainly provide an order-of-magnitude reference.

## D. Conditioning and Stability

Our ISS-style bound depends on the conditioning of the ECF feature map $h(\Phi)$ and its Jacobian $J_h$, whose row norms scale with $\|\omega_\ell\|$. Sampling $\omega \sim \mathcal{N}(0, \Sigma^{-1})$ normalizes the typical phase scale: for pre-normalization descriptors with covariance approximately $\Sigma$, $\mathrm{Var}(\langle\omega, x\rangle) \approx \mathbb{E}[\omega^\top \Sigma \omega] = \mathrm{Tr}(\Sigma^{-1}\Sigma) = D$, which is invariant to anisotropy in $\Sigma$. This mitigates class-to-class variability compared to $\omega \sim \mathcal{N}(0, I)$, where phase variance would scale with $\mathrm{Tr}(\Sigma)$ and can deteriorate local conditioning and step-size bounds.

*Remark on $\ell_2$-normalization.* The descriptors in (7) are $\ell_2$-normalized, which alters the covariance structure. However, the anchor frequencies are sampled using $\Sigma$ computed from *pre-normalization* PCA projections. In practice, the normalization concentrates descriptors on $\mathbb{S}^{D-1}$, and the pre-normalization covariance $\Sigma$ remains a reasonable proxy for controlling phase variance across principal directions.

## D. ECF-Based Gradients vs. Direct MMD Matching

Since the objectives are equivalent (Section A), why not directly minimize the Gaussian-kernel MMD? Two properties favor the ECF/random-feature form in the noisy-$\hat{x}_0$ setting:

*(i) Low-rank, globally-coupled gradient structure.* Let $\Omega = [\omega_1, \ldots, \omega_{K_f}] \in \mathbb{R}^{D \times K_f}$ collect the anchor frequencies. The ECF gradient for all $K_c$ samples can be written as $\widehat{G} = \Omega C^\top$, where $C \in \mathbb{R}^{K_c \times K_f}$ contains bounded trigonometric coefficients. Hence $\mathrm{rank}(\widehat{G}) \leq \min(D, K_f)$: all samples share the *same* frequency directions $\{\omega_\ell\}$, and a common residual $(a_\ell, b_\ell)$ simultaneously steers every sample. This yields a coherent, low-rank "global alignment push." In contrast, direct MMD gradients couple samples via pairwise kernel weights $k(x_i, x_j)$, forming a high-rank, locally-structured interaction graph whose effective rank scales with $K_c$.

*(ii) Noise robustness.* Let $\tilde{x} = x + \varepsilon$ with $\varepsilon \sim \mathcal{N}(0, \sigma^2 I)$. For ECF, noise induces phase jitter: $\mathbb{E}[e^{i\omega^\top \tilde{x}}] = e^{i\omega^\top x} \exp(-\tfrac{1}{2}\sigma^2 \|\omega\|^2)$. High-frequency components are exponentially damped in expectation, acting as an *implicit low-pass filter*; moreover, $\sin/\cos$ are bounded, so instantaneous gradient magnitudes remain controlled. For direct MMD, noise enters the exponent of the kernel: $k_\Lambda(\tilde{x}, \tilde{y}) = \exp(-\tfrac{1}{2}\|\tilde{x} - \tilde{y}\|_\Lambda^2)$. This "exponential-of-quadratic" coupling amplifies noise multiplicatively—pairs with large true distance $\|x - y\|$ yield near-zero kernel values (gradient vanishing), while close pairs exhibit high variance, degrading SNR especially in high dimensions.

*Quantitative comparison.* Under isotropic noise $\varepsilon \sim \mathcal{N}(0, \sigma^2 I)$, the gradient variance scales as:

- **ECF:** $\mathrm{Var}(\nabla_\phi \hat{C}(\omega)) = O(\|\omega\|^2)$, bounded by anchor norm selection.

- **Direct MMD:** $\mathrm{Var}(\nabla_\phi k_\Lambda(\phi, \phi')) = O(\|\phi - \phi'\|_\Lambda^2 \cdot e^{-\|\phi - \phi'\|_\Lambda^2})$, which exhibits high variance for close pairs and vanishing gradients for distant pairs.

Hence ECF-based matching offers more stable gradients when operating on noisy predicted latents $\hat{x}_0$.

### A.4. Proofs for Section 4.4

**Closed-Form Gradients.** We provide closed-form expressions for the descriptor-space directions used in SCG.

**Kernel coupling.** Recall $V_{\mathrm{cpl}}$ from (12) with $\kappa(a, b) = \exp(-\|a - b\|^2/(h + \varepsilon))$. For fixed $k$, using $\nabla_{\phi^{(k)}} \kappa(\phi^{(k)}, \phi^{(j)}) = -\frac{2}{h+\varepsilon} \kappa_{kj}(\phi^{(k)} - \phi^{(j)})$ and symmetry of the double sum, one obtains

$$-\nabla_{\phi_n^{(k)}} V_{\mathrm{cpl}}(\Phi_n) = \frac{2}{(h + \varepsilon)K_c} \Big( \sum_j \kappa_{kj} \phi_n^{(j)} - \phi_n^{(k)} \sum_j \kappa_{kj} \Big).$$

This matches the direction in (13).

**Spectral regulation.** For $\omega_\ell$, define $\hat{C}_\ell = \hat{C}(\omega_\ell)$ and write the regulation residuals $a_\ell = \mathrm{Re}\,\hat{C}_\ell - \mathrm{Re}\,\hat{C}_\ell^\star$ and $b_\ell = \mathrm{Im}\,\hat{C}_\ell - \mathrm{Im}\,\hat{C}_\ell^\star$. Let $\theta_{k\ell} = \langle \omega_\ell, \phi_n^{(k)} \rangle$. Then $V_{\mathrm{reg}} = \sum_{\ell=1}^{K_f}(a_\ell^2 + b_\ell^2)$ and a direct calculation yields

$$s_n^{(k)} = -\nabla_{\phi_n^{(k)}} V_{\mathrm{reg}}(\Phi_n) = \frac{2}{K_c} \sum_{\ell=1}^{K_f} \Big( a_\ell \sin(\theta_{k\ell}) - b_\ell \cos(\theta_{k\ell}) \Big) \omega_\ell.$$

**Proof of Lemma 4.1 (Separability Implies Additivity).** Assume $U$ is continuously differentiable and that for each $k$, the partial gradient satisfies $\nabla_{\phi^{(k)}} U(\Phi) = -G^{(k)}(\Phi) = -g_k(\phi^{(k)})$, i.e., it depends only on $\phi^{(k)}$. Fix any index $k$ and integrate with respect to $\phi^{(k)}$ while holding the other blocks $\phi^{(j)}$ ($j \neq k$) fixed:

$$U(\Phi) = \psi_k(\phi^{(k)}) + C_k(\phi^{(1)}, \ldots, \phi^{(k-1)}, \phi^{(k+1)}, \ldots, \phi^{(K_c)}),$$

for some antiderivative $\psi_k$ and some residual term $C_k$ that does not depend on $\phi^{(k)}$. Applying the same argument sequentially for $k = 1, \ldots, K_c$ yields

$$U(\Phi) = \sum_{k=1}^{K_c} \psi_k(\phi^{(k)}) + \mathrm{const},$$

so $U$ contains no cross terms and is additive across samples.

**Proof of Proposition 4.2 (Lyapunov Decrease).** Assume $V$ is $L$-smooth on $\mathcal{M} = (\mathbb{S}^{D-1})^{K_c}$ under the normalization retraction $\Pi_\mathcal{M}$ (equivalently, it satisfies a retraction-based descent lemma) (Absil et al., 2008), and abstract one SCG injection step as

$$\tilde{\Phi}_{n+1} = \Phi_n + \eta u_n, \qquad \Phi_{n+1} = \Pi_\mathcal{M}(\tilde{\Phi}_{n+1}), \qquad u_n = -\nabla_\mathcal{M} V(\Phi_n) + \Delta_n, \tag{24}$$

where $\Delta_n$ aggregates mismatch induced by backbone evolution/noise, descriptor extraction from $\hat{x}_0$, and the approximate lift $\mathcal{P}$. By the retraction-based descent lemma and (24),

$$V(\Phi_{n+1}) \leq V(\Phi_n) + \eta \langle \nabla_\mathcal{M} V(\Phi_n), u_n \rangle + \frac{L\eta^2}{2} \|u_n\|^2.$$

Substituting $u_n = -\nabla_\mathcal{M} V(\Phi_n) + \Delta_n$ and using $2\langle a, b \rangle \leq \|a\|^2 + \|b\|^2$ and $\|a + b\|^2 \leq 2\|a\|^2 + 2\|b\|^2$ gives

$$V(\Phi_{n+1}) \leq V(\Phi_n) - \eta \|\nabla_\mathcal{M} V(\Phi_n)\|^2 + \eta \langle \nabla_\mathcal{M} V(\Phi_n), \Delta_n \rangle + L\eta^2 \|\nabla_\mathcal{M} V(\Phi_n)\|^2 + L\eta^2 \|\Delta_n\|^2$$

$$\leq V(\Phi_n) - \Big( \frac{\eta}{2} - L\eta^2 \Big) \|\nabla_\mathcal{M} V(\Phi_n)\|^2 + \Big( \frac{\eta}{2} + L\eta^2 \Big) \|\Delta_n\|^2.$$

For $\eta \leq \frac{1}{4L}$, we have $\frac{\eta}{2} - L\eta^2 \geq \frac{\eta}{4}$, hence

$$V(\Phi_{n+1}) \leq V(\Phi_n) - \frac{\eta}{4} \|\nabla_\mathcal{M} V(\Phi_n)\|^2 + c_\Delta \|\Delta_n\|^2,$$

for some $c_\Delta = \frac{\eta}{2} + L\eta^2 = O(\eta)$. If $V$ is lower bounded by $V_{\inf}$ and $\|\Delta_n\| \leq \bar{\Delta}$, then summing the inequality over $n = 0, \ldots, N-1$ yields

$$\frac{\eta}{4} \sum_{n=0}^{N-1} \|\nabla_{\mathcal{M}} V(\Phi_n)\|^2 \leq V(\Phi_0) - V(\Phi_N) + c_\Delta \sum_{n=0}^{N-1} \|\Delta_n\|^2 \leq V(\Phi_0) - V_{\inf} + N c_\Delta \bar{\Delta}^2.$$

Dividing by $N$ and rearranging gives (16) with $C_0 = \frac{4(V(\Phi_0) - V_{\inf})}{\eta}$ and $C_1 = \frac{4c_\Delta}{\eta}$.

**Proof of Proposition 4.3 (ISS for Distributional Tracking).** Let $e_n = h(\Phi_n) - h_c^\star$ and consider the regulation energy $V_{\mathrm{reg}}(\Phi_n) = \|e_n\|_2^2$. Ignoring higher-order terms, a first-order expansion yields

$$e_{n+1} \approx e_n + J_h(\Phi_n)\,(\Phi_{n+1} - \Phi_n),$$

where $J_h = \partial h / \partial \Phi$. Write the injected direction (in the ambient coordinates) as a regulation term plus coupling and mismatch:

$$\Phi_{n+1} - \Phi_n \approx \eta\Big(-\beta\, J_h^\top e_n + \alpha_n\, u_{\mathrm{cpl}}(\Phi_n)\Big) + d_n,$$

where $u_{\mathrm{cpl}}(\Phi_n) \approx -\nabla_{\mathcal{M}} V_{\mathrm{cpl}}(\Phi_n)$ is the coupling direction, we absorb constant factors into $\eta$, and $d_n$ collects the mismatch terms and higher-order truncation (including retraction linearization). This gives an affine recursion

$$e_{n+1} \approx \big(I - \eta\beta\, J_h J_h^\top\big)e_n + \eta\alpha_n\, J_h u_{\mathrm{cpl}}(\Phi_n) + d_n,$$

If $J_h J_h^\top \succeq \mu I_{2K_f}$ locally and $\eta\beta \leq 1/\|J_h\|^2$, then $\|I - \eta\beta J_h J_h^\top\| \leq 1 - \mu\eta\beta$, yielding

$$\|e_{n+1}\| \leq (1 - \mu\eta\beta)\|e_n\| + \eta\alpha_n \|J_h u_{\mathrm{cpl}}(\Phi_n)\| + \|d_n\|.$$

Bounding $\|J_h u_{\mathrm{cpl}}(\Phi_n)\| \leq C_{\mathrm{cpl}}$ locally and bounding $\|d_n\|$ linearly by $\epsilon(t_n)$ and $\epsilon_{\mathcal{P}}(t_n)$ produces (17) (up to constants). Moreover, iterating the recursion gives $\|e_n\| \leq (1 - \mu\eta\beta)^n \|e_0\| + \sum_{k=0}^{n-1}(1 - \mu\eta\beta)^{n-1-k}\big(\eta\alpha_k C_{\mathrm{cpl}} + \|d_k\|\big)$, so $\|e_n\| \to 0$ when $\alpha_n \to 0$ and $\|d_n\| \to 0$.

**A Sufficient Step-Size Bound for the ECF Map.** From (9), each row of $J_h$ corresponds to $\nabla_{\phi^{(k)}} \mathrm{Re}\, \hat{C}(\omega_\ell)$ or $\nabla_{\phi^{(k)}} \mathrm{Im}\, \hat{C}(\omega_\ell)$, and satisfies $\|\nabla_{\phi^{(k)}} \mathrm{Re}\, \hat{C}(\omega_\ell)\| \leq \|\omega_\ell\|/K_c$ and $\|\nabla_{\phi^{(k)}} \mathrm{Im}\, \hat{C}(\omega_\ell)\| \leq \|\omega_\ell\|/K_c$. Each row concatenates $K_c$ such blocks, so its norm satisfies $\|\mathrm{row}\| \leq \|\omega_\ell\|/\sqrt{K_c}$. Therefore $\|J_h\| \leq \sqrt{2K_f}\,\omega_{\max}/\sqrt{K_c}$ where $\omega_{\max} = \max_\ell \|\omega_\ell\|$, and it suffices to choose $\eta\beta \leq K_c/(2K_f\,\omega_{\max}^2)$ (equivalently, $\eta \leq K_c/(2\beta K_f\,\omega_{\max}^2)$) to ensure $\eta\beta \leq 1/\|J_h\|^2$.

**Proof of Proposition 4.4 (Anti-Collapse).** If $\phi_n^{(1)} = \cdots = \phi_n^{(K_c)} = \bar{\phi}$, then $\hat{C}(\omega) = \frac{1}{K_c}\sum_j e^{i\langle\omega,\bar{\phi}\rangle} = e^{i\langle\omega,\bar{\phi}\rangle}$ and hence $|\hat{C}(\omega)| = 1$ for all $\omega$. If there exists an anchor $\omega_\ell$ such that $|\hat{C}^\star(\omega_\ell)| \leq 1 - \rho$, then by the reverse triangle inequality, $|\hat{C}(\omega_\ell) - \hat{C}^\star(\omega_\ell)| \geq \big||\hat{C}(\omega_\ell)| - |\hat{C}^\star(\omega_\ell)|\big| \geq \rho$. Identifying $|\hat{C}(\omega_\ell) - \hat{C}^\star(\omega_\ell)|$ with the $\ell$-th two-dimensional residual norm in $h(\Phi)$ yields

$$\big\|[\mathrm{Re}\,\hat{C}(\omega_\ell), \mathrm{Im}\,\hat{C}(\omega_\ell)] - [\mathrm{Re}\,\hat{C}^\star(\omega_\ell), \mathrm{Im}\,\hat{C}^\star(\omega_\ell)]\big\|_2 \geq \rho,$$

which implies $V_{\mathrm{reg}}(\Phi_n) = \|h(\Phi_n) - h_c^\star\|_2^2 \geq \rho^2$.

## A.5. Stabilization Heuristics Mentioned in the Method

Our implementation uses two lightweight stabilizers: (i) time-dependent gain scheduling/gating, and (ii) input saturation via RMS clipping (automatic gain control) to bound the injected feedback magnitude.

**Time-Dependent Gain Scheduling and Gating.** Let $T$ be the total number of diffusion steps and let $t_n$ denote the reverse timestep at iteration $n$. Define the normalized time $\tau = t_n/T \in [0, 1]$. We use fixed schedules (no tuning) consistent with our implementation:

$$\gamma(t_n) = \tau, \qquad m(t_n) = \mathbf{1}\{0.3 \leq \tau \leq 0.7\}, \qquad s(t_n) = s_{\min} + (1 - \tau)^2(s_{\max} - s_{\min}),$$

where $m(t_n)$ gates the spectral regulation term to a mid-timestep window, and $s(t_n)$ rescales frequency anchors as $\omega_\ell(t_n) = s(t_n)\omega_\ell$ to progressively focus from low to high frequencies. We use $s_{\min} = 0.5$ and $s_{\max} = 2.0$. In practice, we treat $\alpha_n = \alpha\,\gamma(t_n)$ and $\beta_n = \beta\,m(t_n)$ as effective time-varying gains multiplying the coupling and regulation directions, respectively.

**RMS Clipping (Automatic Gain Control).** Let $g_{\text{base}}$ denote the stacked base directions (same tensor shape as the injected term) and let $g_{\text{tot}} = g_{\text{base}} + u$ be the total injected direction (base plus SCG controller). We compute a global root-mean-square magnitude

$$\text{RMS}(G) = \sqrt{\tfrac{1}{|G|} \sum_{e \in G} e^2}.$$

We set a reference magnitude

$$\text{RMS}_{\text{ref}} = \begin{cases} \text{RMS}(g_{\text{base}}), & \text{RMS}(g_{\text{base}}) \geq \varepsilon, \\ \text{RMS}(\hat{x}_0) \text{ or } \text{RMS}(z_t), & \text{otherwise}, \end{cases}$$

with $\varepsilon = 10^{-6}$ to handle settings where the base direction is (near) zero (e.g., vanilla sampling). We then apply hard saturation

$$g_{\text{tot}} \leftarrow g_{\text{tot}} \cdot \min\Big(1, \frac{r_{\max} \text{RMS}_{\text{ref}}}{\text{RMS}(g_{\text{tot}})}\Big),$$

where we set $r_{\max} = 2.0$ in all experiments.

**Kernel Bandwidth (Median Heuristic).** At each timestep, we recompute the RBF bandwidth $h$ from the current within-class descriptors $\{\phi_n^{(k)}\}_{k=1}^{K_c}$ (and treat it as a constant w.r.t. gradients). Let $d_{ij}^2 = \|\phi_n^{(i)} - \phi_n^{(j)}\|_2^2$ and let $\mathcal{I} = \{(i,j) : i \neq j\}$. We set

$$h = \max\Big(10^{-6}, \text{median}_{(i,j)\in\mathcal{I}} d_{ij}^2 \cdot \rho_{\text{bw}}/\log(K_c + 1)\Big),$$

with $\rho_{\text{bw}} = 1.0$; if $\mathcal{I}$ is empty, we set $h = 1$.

**Controller Complexity.** Per reverse step and per class, the SCG controller computes: (i) descriptor projection via PCA, $\phi_n^{(k)} = \text{normalize}(W_c^\top(\text{vec}(\hat{x}_{0,n}^{(k)}) - \mu_c))$, costing $O(K_c d_{\text{full}} D)$; (ii) ECF features and their gradients over $K_f$ anchors, costing $O(K_c K_f D)$; and (iii) kernel coupling, costing $O(K_c^2 D)$ for pairwise similarities. These costs are typically small compared to a diffusion-backbone forward pass, so in Table 4 they do not materially change the reported GFLOPs at the shown precision.

### A.6. Storage Overhead

SCG does not introduce additional storage in the final distilled dataset: after sampling, only the distilled images are kept. The controller itself requires a small per-class reference bundle computed once offline from real data: the PCA basis $(\mu_c, W_c)$ for descriptor extraction (Equation (7)), the anchor frequencies $\{\omega_\ell\}_{\ell=1}^{K_f}$, and the target ECF features $h_c^\star$ (Equation (9)). If cached, this bundle uses $O(d_{\text{full}} D + d_{\text{full}} + K_f D + K_f)$ floating-point numbers per class, and can be discarded after finishing class $c$ (so peak overhead is per-class and independent of the number of diffusion steps). For large-scale settings such as ImageNet-1K, the reference bundle can be computed and consumed in a streaming manner over small class batches (e.g., 10 classes at a time) and discarded immediately, keeping the peak storage overhead low in practice. In our DiT setup, sampling is performed in VAE latent space, so $d_{\text{full}}$ is the flattened VAE latent size. With an SD-VAE-style latent (4 channels, downsample factor 8), an image of size $S \times S$ has latent size $4 \cdot (S/8) \cdot (S/8)$. For $S = 256$ ($d_{\text{full}} = 4 \cdot 32 \cdot 32 = 4096$) with $D = 64$ and $K_f = 32$, the per-class bundle is about $1.0$ MB if stored as raw FP32 (half in FP16). On disk, the exact size depends on serialization and optional compression; for dense floating-point arrays, the dominant lever is precision (FP16 vs. FP32), while compression typically yields a smaller additional gain.

## B. Additional Experiments

### B.1. Diversity/Collapse Metrics

We compute diversity/collapse diagnostics on the within-class descriptor set $\Phi = \{\phi^{(k)}\}_{k=1}^{K_c}$ extracted from predicted clean latents (normalized as in (7)). Let $\bar{\phi} = \frac{1}{K_c} \sum_k \phi^{(k)}$ and $\Sigma = \frac{1}{K_c} \sum_k (\phi^{(k)} - \bar{\phi})(\phi^{(k)} - \bar{\phi})^\top$. The mean nearest-neighbor distance is

$$d_{\text{NN}} = \frac{1}{K_c} \sum_{k=1}^{K_c} \min_{j \neq k} \|\phi^{(k)} - \phi^{(j)}\|_2,$$

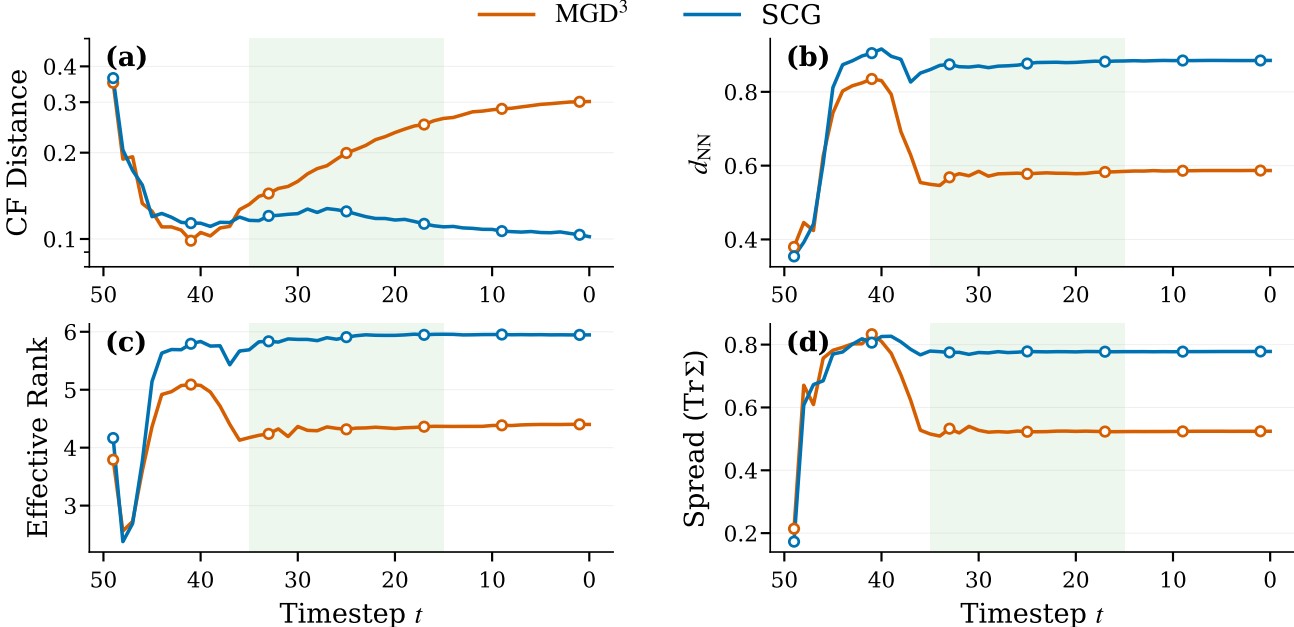

*Figure 3.* Set-level alignment and diversity diagnostics on ImageNette (IPC=10). (a) CF distance (lower is better). (b) Nearest-neighbor distance $d_{\mathrm{NN}}$ (higher is better). (c) Effective rank (higher is better). (d) Spread $\mathrm{Tr}(\Sigma)$ (higher is better). The shaded region marks timesteps where the spectral regulation term is active.

*Table 7.* ImageNette ResNet-18 accuracy (%) across IPC. Additional IPC=5/100/200 runs use the same hard-label evaluation protocol.

| IPC | DiT | DiT+SCG | MGD$^3$ | MGD$^3$+SCG |
|---|---|---|---|---|
| 5 | 51.0 | 51.2 | 53.4 | 57.2 |
| 10 | 58.2 | 65.8 | 61.5 | 64.1 |
| 20 | 62.0 | 69.4 | 69.1 | 73.0 |
| 50 | 71.6 | 79.6 | 78.1 | 82.6 |
| 100 | 73.8 | 75.8 | 83.0 | 84.4 |
| 200 | 76.8 | 78.8 | 87.6 | 88.2 |

the spread is $\mathrm{Tr}(\Sigma)$, and the effective rank is $\mathrm{erank}(\Sigma) = \exp\left(-\sum_i p_i \log p_i\right)$ where $p_i = \lambda_i / \sum_j \lambda_j$ for eigenvalues $\{\lambda_i\}$ of $\Sigma$.

### B.2. Set-Level Alignment and Diversity Diagnostics

Figure 3 reports diagnostic curves on ImageNette (IPC=10): the CF distance minimized by spectral regulation and several within-class diversity metrics (definitions in Section B.1).

### B.3. IPC and Dataset Scaling

Table 7 combines the ImageNette ResNet-18 results from the main experiments with additional IPC=5/100/200 runs. Accuracy increases as IPC grows, but the incremental benefit of SCG is not monotone: for MGD$^3$, the gain is largest at medium IPC (20–50) and tapers at IPC=100/200. This is consistent with a finite-sample interpretation: as $K$ grows, the empirical synthetic set already approximates the target distribution more closely and evaluator accuracy moves closer to saturation, reducing the marginal value of explicit set-level correction.

### B.4. Alternative Set-Level Objectives

The SCG controller is not tied to ECF as the only possible set-level objective. Characteristic-function matching has also been explored for static minmax dataset distillation in NCFM (Wang et al., 2025b); our use is different because the fixed ECF anchors define a closed-form reverse-time controller rather than a learned neural discrepancy in an outer optimization loop.

*Table 8.* Single-class ImageNette sampling benchmark with effective IPC larger than the execution micro-batch.

| IPC | Actual Batch | Peak VRAM (MiB) | Sampling Time (s) | Wall Time (s) |
|-----|--------------|-----------------|-------------------|---------------|
| 50  | 50           | 27432           | 20.27             | 34.05         |
| 100 | 50           | 27432           | 41.85             | 49.33         |
| 200 | 50           | 27473           | 85.62             | 88.16         |

*Table 9.* Long-tail real-data sanity check on ImageNette with balanced IPC=10 output.

| Method | ResNet-18 |
|--------|-----------|
| $MGD^3$ | 62.4 |
| $MGD^3$+SCG | **65.0** |

For example, an RBF-MMD objective provides closed-form descriptor gradients on the current synthetic set, and can be inserted into the same project–feedback–lift controller template. In an ImageNette IPC=10 descriptor-space pilot, replacing the ECF target with RBF-MMD remains stable but gives lower ResNet-18 accuracy than the default ECF controller ($64.6\%$ vs. $65.8\%$). ECF is therefore not claimed as a new discrepancy metric; its practical advantage here is that it compresses the real-data reference into fixed anchor statistics, keeps per-step cost decoupled from real-bank size, and aligns with the tracking analysis in Section 4.4. Wasserstein/OT objectives are also attractive for dataset distillation (Cui et al., 2025; Liu et al., 2025), but using them as an online per-step controller requires solving or approximating a transport plan at each step, whereas the ECF controller uses a fixed real-data statistic.

**FID-Style Objectives.** We also tested a descriptor-space exact-FID pilot. Using FID as an online controller objective requires differentiating through the descriptor mean, covariance, and matrix square root; at IPC=10 this is numerically sensitive because the synthetic covariance is low rank. With $\epsilon I$ regularization, PSD square root via eigendecomposition, and eigenvalue clamping, the exact-FID pilot samples without `nan`/`inf`, but reaches only $62.4\%$ ResNet-18 accuracy on ImageNette IPC=10, below the default ECF setting ($65.8\%$). Its per-class sampling time is similar to ECF (5.127s vs. 5.021s) and peak memory is also similar (about 5.38 GiB), so the extra numerical complexity did not yield an empirical advantage in this setting.

### B.5. Exact Set Sampling for Large IPC

Naively processing all particles of a class in one forward pass can exhaust memory at large IPC. Our implementation therefore uses a three-phase exact set sampling schedule: (i) feature collection, where the diffusion backbone runs on micro-batches and caches predicted clean latents, descriptors, and base guidance for all $K$ particles; (ii) global interaction, where SCG computes ECF regulation and kernel coupling once on the full $K$-particle descriptor bank; and (iii) batched injection, where the precomputed controller is written back to each micro-batch. Only backbone execution is chunked; the set interaction remains global. Thus IPC=200 with micro-batch 50 is one set-coupled reverse step executed in four chunks, not four independent group samplings.

Table 8 shows that peak VRAM remains nearly constant at about 27.4 GiB when actual execution batch is fixed to 50, while runtime scales roughly linearly with effective IPC. This addresses the main OOM risk of group sampling, although large IPC still increases wall-clock cost.

### B.6. Class-Imbalanced Real Descriptor Banks

SCG is applied to a pretrained diffusion prior and uses per-class real descriptors only to compute reference statistics, so it can be used when the real descriptor bank is class-imbalanced. As a sanity check, we construct a long-tail ImageNette real set with imbalance factor 10 (largest/smallest class count about 10:1) while keeping the distilled output balanced at IPC=10. Under ResNet-18 evaluation, $MGD^3$ improves from 62.4 to 65.0 after adding SCG. This preliminary result supports applicability when the real descriptor bank is imbalanced, while training diffusion models from scratch on imbalanced data remains outside our scope.

*Table 10.* IGD compatibility results on ImageNette and ImageWoof. "IGD" uses standard deviation guidance; "IGD(-DEV)" disables it ($\gamma_t = 0$); "IGD(-DEV)+SCG" integrates SCG on top. **Bold**: best in each column.

| Dataset | Method | ConvNet-6 | | ResNetAP-10 | | ResNet-18 | |
|---|---|---|---|---|---|---|---|
| | | 10 | 20 | 10 | 20 | 10 | 20 |
| ImageNette | IGD | $59.9_{\pm 1.0}$ | $65.7_{\pm 1.4}$ | $65.9_{\pm 0.4}$ | $\mathbf{71.8}_{\pm 1.1}$ | $\mathbf{67.7}_{\pm 0.7}$ | $73.0_{\pm 1.4}$ |
| | IGD(-DEV) | $57.8_{\pm 1.2}$ | $63.2_{\pm 1.6}$ | $62.4_{\pm 0.9}$ | $68.8_{\pm 1.3}$ | $61.8_{\pm 1.5}$ | $67.2_{\pm 1.0}$ |
| | IGD(-DEV)+SCG | $\mathbf{63.2}_{\pm 1.0}$ | $\mathbf{67.0}_{\pm 1.3}$ | $\mathbf{67.4}_{\pm 1.2}$ | $71.2_{\pm 1.1}$ | $67.2_{\pm 1.4}$ | $\mathbf{73.4}_{\pm 1.5}$ |
| ImageWoof | IGD | $31.4_{\pm 1.0}$ | $36.6_{\pm 0.6}$ | $38.6_{\pm 1.3}$ | $44.4_{\pm 0.9}$ | $40.4_{\pm 1.3}$ | $\mathbf{50.9}_{\pm 1.2}$ |
| | IGD(-DEV) | $32.4_{\pm 1.3}$ | $33.6_{\pm 1.1}$ | $38.0_{\pm 1.5}$ | $44.0_{\pm 1.2}$ | $39.0_{\pm 1.4}$ | $44.8_{\pm 1.6}$ |
| | IGD(-DEV)+SCG | $\mathbf{33.8}_{\pm 1.4}$ | $\mathbf{37.0}_{\pm 1.2}$ | $\mathbf{41.6}_{\pm 1.0}$ | $\mathbf{45.2}_{\pm 1.5}$ | $\mathbf{45.0}_{\pm 1.1}$ | $45.8_{\pm 1.7}$ |

## B.7. Compatibility with IGD

IGD (Chen et al., 2025) augments guided sampling with a deviation (DEV) guidance term that explicitly depends on a per-class memory $\mathcal{M}_c$ of previously generated samples. Concretely, when generating the $k$-th sample, the DEV term repels the current trajectory away from previously generated samples $\{z^{(1)}, \ldots, z^{(k-1)}\}$, encouraging within-class diversity through a sequential, order-dependent mechanism.

**Structural Incompatibility.** The DEV term induces a *causal* (lower-triangular) coupling: sample $k$ depends on samples $1, \ldots, k-1$, but not vice versa. In contrast, SCG implements a *dense* (symmetric) coupling: every sample's update depends on the entire within-class state $\mathcal{Z}^c$ simultaneously. These two coupling structures are fundamentally incompatible in a single forward pass: the DEV term requires a fixed generation order with incrementally updated memory, while SCG requires all $K_c$ samples to be available and updated jointly at each step. Attempting to combine both terms without additional ordering-specific engineering leads to unstable generation, as the causal DEV gradients conflict with the symmetric SCG feedback.

**Integration Strategy.** To integrate SCG with IGD, we disable the deviation guidance (setting $\gamma_t = 0$) and rely on SCG to provide the diversity mechanism. This raises a natural question: can SCG's set-symmetric coordination recover or exceed the diversity benefits of the sequential DEV term? In IGD(-DEV)+SCG, the influence guidance of IGD is kept, the DEV guidance is disabled, and the full SCG controller (both $V_{\text{reg}}$ and $V_{\text{cpl}}$) is added. Thus Table 10 tests whether the full set-coupled controller can replace the sequential DEV mechanism; it is not intended to isolate coupling alone.

**Experimental Results.** Table 10 reports results on ImageNette and ImageWoof at IPC$\in \{10, 20\}$. We compare three configurations: (i) standard IGD with DEV enabled, (ii) IGD(-DEV) with deviation guidance disabled, and (iii) IGD(-DEV)+SCG where SCG replaces the DEV term.

Disabling DEV (row 2 vs. row 1) consistently degrades performance, confirming that the deviation term provides meaningful diversity benefits—e.g., on ImageNette (IPC=10), ResNet-18 accuracy drops from 67.7% to 61.8% ($-5.9\%$). Adding SCG (row 3 vs. row 2) recovers and often exceeds the original IGD performance. On ImageNette, IGD(-DEV)+SCG matches or surpasses full IGD across all architectures and IPC settings (e.g., ResNetAP-10 at IPC=10: 67.4% vs. 65.9%, $+1.5\%$). On the harder ImageWoof benchmark, SCG yields substantial gains over IGD(-DEV) and approaches or exceeds full IGD in most settings (e.g., ResNet-18 at IPC=10: 45.0% vs. 40.4%, $+4.6\%$).

Within the tested IPC range (10–20), these results suggest that SCG can serve as an effective alternative to the DEV term, recovering comparable or improved accuracy while enabling IPC-at-once generation. Whether this trade-off remains favorable at higher IPC budgets is left for future work.

## C. Image Visualization

### C.1. Single-Class t-SNE Visualization

We visualize single-class t-SNE embeddings of VAE features for ImageNette class `n03000684` and ImageWoof class `n02096294` (IPC=10). In each plot, blue dots denote real samples (with KDE contours); crosses/circles denote baseline/SCG-augmented distilled samples, respectively.

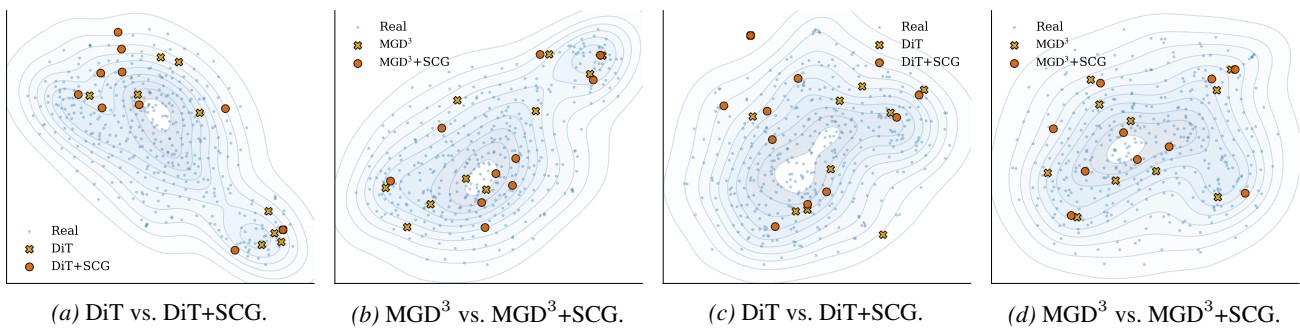

*(a) DiT vs. DiT+SCG.*     *(b) MGD$^3$ vs. MGD$^3$+SCG.*     *(c) DiT vs. DiT+SCG.*     *(d) MGD$^3$ vs. MGD$^3$+SCG.*

*Figure 4.* Single-class t-SNE (IPC=10). ImageNette: class `n03000684` (left two). ImageWoof: class `n02096294` (right two).

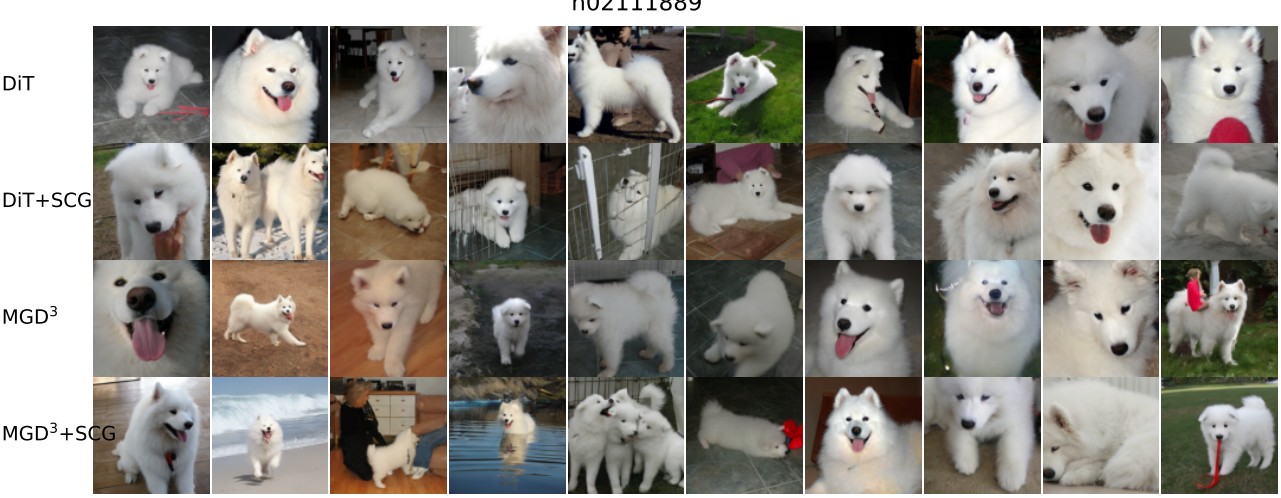

*Figure 5.* Qualitative comparison on ImageWoof class `n02111889` (IPC=10).

**RBF-MMD and Computation.** We quantify real–distilled distribution alignment in the original feature space using the squared MMD with an RBF kernel $k(a, b) = \exp(-\|a - b\|_2^2/(2\sigma^2))$. Given real features $X = \{x_i\}_{i=1}^n$ and distilled features $Y = \{y_j\}_{j=1}^m$, we compute the (biased) estimate

$$\text{MMD}^2(X, Y) = \frac{1}{n^2} \sum_{i,i'} k(x_i, x_{i'}) + \frac{1}{m^2} \sum_{j,j'} k(y_j, y_{j'}) - \frac{2}{nm} \sum_{i,j} k(x_i, y_j).$$

We set $\sigma^2$ via a median heuristic on pairwise squared distances within $X$; lower MMD indicates closer distributions. This estimator requires $O(n^2 + m^2 + nm)$ kernel evaluations and can be computed efficiently with matrix operations. For the selected classes, SCG reduces MMD for both backbones: ImageNette `n03000684` 0.0829 $\rightarrow$ 0.0393 (DiT) and 0.0463 $\rightarrow$ 0.0358 (MGD$^3$); ImageWoof `n02096294` 0.0467 $\rightarrow$ 0.0411 (DiT) and 0.0383 $\rightarrow$ 0.0339 (MGD$^3$).

### C.2. Qualitative Comparison

Figure 5 shows distilled images for ImageWoof class `n02111889` (IPC=10) across four methods. SCG-augmented variants exhibit greater visual diversity while maintaining class consistency.

