# OpenReview forum: "Set-Coupled Guidance: Set-Level Coordination in Diffusion-Based Dataset Distillation"
_ICML.cc/2026/Conference — ICML 2026 regular_

### Official Review · Reviewer_svYS · 2026-02-24

**Soundness:** 3
**Presentation:** 3
**Significance:** 3
**Originality:** 3
**Overall Recommendation:** 4
**Confidence:** 4

**Summary:**

The authors propose SCG, a plug-and-play auxiliary controller that reformulates the independent, image-by-image sampling paradigm typically used in DD into a set-level "IPC-at-once" approach. Specifically, the authors inject set-symmetric feedback representing the entire in-class sample state into each diffusion step, thereby enabling co-optimization across samples. This methodology is implemented via two primary components: Spectral Set-Point Regulation and Cooperative Kernel Coupling.

**Compliance With Llm Reviewing Policy:**

Affirmed.

**Final Justification:**

The author addressed my concerns and questions

**Key Questions For Authors:**

No.

**Limitations:**

Yes.

**Strengths And Weaknesses:**

Strengths

1. The authors propose a novel shift from traditional per-sample update rules to a group-based (IPC-at-once) sampling framework. By injecting set-symmetric feedback, the authors effectively address set-level redundancy that independent sampling fails to detect.

2. The authors provide a rigorous theoretical analysis, including Lyapunov-based convergence and ISS for distributional tracking. Additionally, the anti-collapse property established for non-degenerate real data offers a principled guarantee against mode collapse.

3. The authors achieve high efficiency, maintaining minimal overhead (typically <2%) despite the inherent complexity of set-level coordination. This is accomplished by computing feedback in a low-dimensional descriptor space via fixed PCA, thereby avoiding costly backpropagation through the diffusion backbone.

4. The authors demonstrate that SCG is highly adaptable and can be seamlessly integrated into various existing diffusion-based DD pipelines, such as $MGD^{3}$ and $CaO_{2}$, without requiring retraining of the generative models.

Weaknesses

1. The authors utilize sophisticated mathematical concepts, such as spectral set-point regulation via ECF matching and cooperative kernel coupling, to ensure diversity and stability. While technically sound, the interplay between these components may be difficult to grasp intuitively. The authors should provide a detailed step-by-step algorithmic flowchart and are encouraged to release the source code to facilitate community understanding and reproducibility.

2. The authors utilize ECF features to align synthetic and real descriptor distributions. However, given that FID remains the gold standard for measuring distributional differences in generative modeling, it is unclear whether FID could be directly employed as an optimization objective within this set-coupled framework. The authors should discuss the potential results, gradient stability, and performance trade-offs of using FID-based optimization compared to the proposed ECF matching.

3. The algorithm requires that all images for a specific class be processed in a single forward pass to compute the symmetric feedback. Intuitively, a larger batch size might lead to a more accurate approximation of the original dataset distribution. The authors should provide experiments showing the performance trend as the batch size increases: is a larger batch size always better, or is there a point of diminishing returns? Furthermore, if larger batch sizes are indeed beneficial, can the authors propose a modification (e.g., via gradient accumulation or streaming) to simulate extremely large batches without triggering Out-of-Memory (OOM) errors?

---

> ### Author Rebuttal · Authors · 2026-03-30
>
> Response to Reviewer svYS
>
> Thank you for your constructive feedback and for recognizing the novelty of our group-based sampling framework, the rigor of our theoretical analysis, and the high efficiency of our method. Below we respond to the three concrete weaknesses raised in the review.
>
> **Q1: Algorithmic clarity and code release**
>
> **A1:** We will release the complete source code upon publication. We will also provide a more detailed algorithm description and clearer pseudocode in the revised manuscript to make the interaction between Spectral Set-Point Regulation and Cooperative Kernel Coupling easier to follow.
>
> **Q2: ECF vs. FID-style objectives**
>
> **A2:** We additionally tested a descriptor-space exact-FID pilot. This objective requires differentiating through the descriptor means, covariances, and matrix square root, so to make it numerically feasible at IPC=10 we used conservative epsilon-identity regularization, a PSD square root via eigendecomposition, and eigenvalue clamping. On ImageNette (IPC=10), this regularized exact-FID pilot samples stably without `nan/inf`, reaches 62.4% ResNet-18 accuracy, uses about 5.38 GiB peak memory, and has similar per-class sampling time to ECF (5.127s vs. 5.021s). However, it still underperforms our main ECF setting (65.8%). Therefore, FID-style objectives are feasible, but they require a more complex differentiable implementation and stronger numerical regularization, while still performing worse than ECF in our setting.
>
> **Q3: Batch scaling and OOM mitigation**
>
> **A3:** Intuitively, a larger IPC gives a better empirical approximation of the target class distribution, so increasing IPC can indeed help set-level generation. Our ImageNette results support this view: the absolute performance rises with IPC, while the *incremental gain from SCG* is strongest at medium IPC (20--50) and then decreases at IPC=100/200. This is consistent with finite-sample statistics: once the empirical set itself becomes closer to the target distribution and evaluator accuracy moves closer to its upper ceiling, the marginal benefit of explicit set-level coordination should taper.
>
> The key intuition behind our memory-saving implementation is that the expensive part is the diffusion backbone forward pass, whereas the set interaction itself is computed only on low-dimensional PCA descriptors. We therefore do **not** split the set-level interaction into independent batches. Instead, we keep the interaction global and split only the backbone execution.
>
> Concretely, we use a **three-phase exact set sampling** strategy:
> (i) **Feature Collection:** run the diffusion backbone on micro-batches and cache the predicted clean latents, descriptor features, and base guidance for all $K$ particles;
> (ii) **Global Interaction:** after all particles have been collected, compute the ECF regulation term, kernel coupling term, and final controller update once on the full $K$-particle descriptor bank;
> (iii) **Batched Injection:** write the precomputed controller back to each micro-batch to complete the same full-set update.
> This is mathematically equivalent to direct IPC-at-once sampling, up to floating-point / execution-order effects, because only the execution schedule changes while the underlying set-level interaction remains global.
>
> In our single-class sampling benchmark, for effective batch sizes of 50, 100, and 200, peak VRAM stays at about 27.4 GB with a micro-batch of 50, while sampling time and total GFLOPs scale approximately linearly with IPC; on our RTX 5090 this does not trigger OOM.
> For the full multi-IPC accuracy trend, please see our response to Reviewer XipB; for the detailed large-IPC sampling cost benchmark, please see our response to Reviewer fzoG.

---

> > ### Author Rebuttal · Reviewer_svYS · 2026-04-01
> >
> > The author addressed my concerns and questions.

---

> > > ### Author Response · Authors · 2026-04-03
> > >
> > > Thank you for the follow-up and for confirming that our rebuttal addressed your concerns. We appreciate your time and consideration.

---

### Official Review · Reviewer_fzoG · 2026-03-05

**Soundness:** 3
**Presentation:** 3
**Significance:** 3
**Originality:** 3
**Overall Recommendation:** 4
**Confidence:** 5

**Summary:**

This paper proposes Set-Coupled Guidance (SCG), analyzes for the first time the limitations of diffusion-based DD methods in terms of set-symmetric feedback at each diffusion step, and proposes group-based sampling with spectral set-point regulation and cooperative kernel coupling. Extensive experiments validate its effectiveness.

**Compliance With Llm Reviewing Policy:**

Affirmed.

**Final Justification:**

SCG is a novel method, and the author's response addressed my main concerns, prompting me to maintain my rating while raising the confidence level to 5.

**Key Questions For Authors:**

1. In Table 6, Does IGD(-DEV)+SCG mean that the diversity guidance is not used, but the influence guidanceis still used, combined with the result of SCG?

2. Regarding the PCA processing described in lines 201-214. Can you provide information on the impact of retaining different principal components on performance and efficiency?

3. Regarding the memory usage of group sampling. The disadvantage of group sampling is obviously that it takes up a lot of memory for independent sampling. Limitations and future work or appendi should be discussed rather than avoided.

**Limitations:**

The memory usage of group sampling, especially when IPC > 50, is likely to cause memory outages on a single NVIDIA RTX 5090 GPU.

**Strengths And Weaknesses:**

**Strengths:**

1. It is insightful to use the state space equation of modern control theory to describe the sample preocess of guided diffusion. The experimental and theoretical analyses are quite thorough.

2. The paper is well written and provides a sufficient analysis and comparison of existing methods.

3. SCG  improves performance at a negligible additional computational cost, which adopts the previous training-free diffusion paradigm.

4. SCG can be applied as a plugin to existing fusion-based methods.


**Weakness**

1. Although group-based sampling is intresting, the proposed solution or optimization objective in the article merely involves adding a distribution-matching guide during the sampling process. If this is the case, methods like DM [1] and MTT [2] would also be applicable. It is regrettable that no diffusion-specific sampling objective was designed.

2. Following up on point 1, the proposed Spectral Set-Point Regulation adopts the neural characteristic function from NCFM [3], which is not novel and lacks proper citation of the original source. This is undesirable.

3. I still have some concerns about the optimization objective. Since DiT itself is trained on ImageNet, there's no need to worry about distribution mismatch. As shown in Figure 4, the synthetic data all fall within the range of the real data distribution. However, the reason for influencing the diffusion trajectory solely through the distribution matching paradigm isn't very clear. The CF distance shown in Figure 3 alone isn't very convincing. Therefore, I suspect the performance improvement comes from fine-tuning multiple hyperparameters and the characteristic functions previously proposed for NCFM. Of course, I don't deny the contribution of this paper to sampling stability. However, why group sampling enhances existing diffusion-based methods is worth discussing in depth. For example, group sampling strengthens the consistency of MGD3, making the embedded images of the synthesized images closer to the prototype, etc.

[1] Dataset Condensation with Distribution Matching. WACV 2023.
[2] Dataset Distillation by Matching Training Trajectories. CVPR 2022.
[3] Dataset Distillation with Neural Characteristic Function: A Minmax Perspective. CVPR 2025.

---

> ### Author Rebuttal · Authors · 2026-03-30
>
> Response to Reviewer fzoG
>
> Thank you for the constructive review.
>
> **Q1: Can you provide information on the impact of retaining different principal components on performance and efficiency?**
>
> **A1:** We ablated PCA dimension $D$ on ImageNette (IPC=10, $K_f=32$, ResNet-18).
>
> | $D$ | Explained Var Ratio | Recon MSE | Top-1 Acc (%) | Sample Time (s) | Peak Mem (MiB) |
> |---:|---:|---:|---:|---:|---:|
> | 16 | 0.3838 | 0.4093 | 65.6 | 59.30 | 5372.8 |
> | 32 | 0.4777 | 0.3469 | 66.2 | 59.28 | 5373.3 |
> | 64 | 0.6018 | 0.2643 | 65.8 | 60.39 | 5374.4 |
> | 128 | 0.7789 | 0.1467 | 66.2 | 62.22 | 5376.5 |
> | 256 | 1.0000 | 0.0000 | 65.4 | 58.68 | 5381.6 |
>
> As $D$ increases, reconstruction MSE decreases monotonically while accuracy stays stable (65.4%--66.2%). Time and memory barely change because DiT/VAE remain the bottleneck, so a moderate rank already works well.
>
> **Q2: In Table 6, does IGD(-DEV)+SCG mean that the diversity guidance is not used, but the influence guidance is still used, combined with SCG?**
>
> **A2:** Yes. In `IGD(-DEV)+SCG`, the DEV (deviation/diversity) guidance is disabled, the influence guidance is kept, and SCG replaces the DEV term.
>
> **Q3: The memory usage of group sampling, especially when IPC > 50, is likely to cause memory outages.**
>
> **A3:** We agree that naive group sampling scales poorly. To address this, we use **three-phase exact set sampling** (Feature Collection $\to$ Global Interaction $\to$ Batched Injection). In a single-class sampling benchmark on ImageNette that isolates one IPC-sized class set at a time, with execution micro-batch fixed at 50, we can scale the effective IPC to 200 without removing any cross-sample interaction, and this runs without OOM on RTX 5090.
>
> Crucially, this remains equivalent to direct IPC-at-once sampling up to floating-point / execution-order effects: full-set interaction is still computed once on the complete feature bank, and the 3-phase scheme changes only the execution schedule.
>
> | IPC | Actual Batch | Peak VRAM (MiB) | Sampling Time (s) | Total GFLOPs |
> |---:|---:|---:|---:|---:|
> | 50 | 50 | 27432 | 20.27 | 4905188.72 |
> | 100 | 50 | 27432 | 41.85 | 9810377.44 |
> | 200 | 50 | 27473 | 85.62 | 19620754.88 |
>
> In this single-class benchmark, peak VRAM stays nearly constant at about 27.4 GB, while sampling time and total GFLOPs scale approximately linearly with IPC. More details are given in our response to Reviewer svYS. We will add this exact-set implementation discussion and the large-IPC results to the revised manuscript.
>
> **Response to weaknesses:**
> *   **Why not simply use DM/MTT-style distribution matching?** DM/MTT-style methods optimize a persistent synthetic dataset in a static outer loop. Our setting is different: SCG acts inside the noisy reverse diffusion trajectory and therefore needs a guidance signal that can be computed analytically at each timestep from the current predicted clean latents of the whole set. The key point is not merely to add a matching objective, but to derive a closed-form, permutation-equivariant guidance law from unordered set statistics that can be injected step-by-step and remain stable under stochastic denoising.
> *   **Missing NCFM citation:** We agree that NCFM (CVPR 2025) should be cited and discussed. However, the overlap is only that both use empirical characteristic functions (ECF). In NCFM, the discrepancy metric itself is made neural: an auxiliary network learns the frequency-sampling strategy in a static min-max outer loop over a persistent synthetic dataset. Our method does not use any such neural parameterization. We use fixed, non-learnable ECF anchors, compute ECF features directly from real/synthetic descriptors, and derive a closed-form per-step permutation-equivariant guidance from the resulting residual, injected into reverse diffusion. Thus, we do not adopt NCFM; we use ECF in a fundamentally different way.
> *   **Why does this influence the trajectory effectively?** A pretrained DiT gives a strong image prior, but a finite set of $K$ jointly generated samples still has an empirical descriptor distribution $\hat P_K$ that generally differs from the target class distribution $P_c$. This is a standard finite-sample effect: empirical means/covariances, or our ECF features $h(\Phi)$, fluctuate around their population values with nonzero error, typically on the order of $O_p(K^{-1/2})$. At small or moderate IPC, these fluctuations are still substantial, so there is room for set-level correction beyond per-sample denoising. SCG does exactly this by feeding back the current ECF residual $h(\Phi_n)-h_c^\star$ and regularizing the joint evolution through kernel coupling. Consistently, at IPC=50 the 10-class average RBF-MMD decreases from $0.00983 \pm 0.00282$ to $0.00958 \pm 0.00156$ on ImageNette and from $0.01129 \pm 0.00310$ to $0.00988 \pm 0.00155$ on ImageWoof after adding SCG to MGD$^3$, supporting correction of finite-sample set mismatch rather than mere hyperparameter tuning.

---

> > ### Author Rebuttal · Reviewer_fzoG · 2026-04-01
> >
> > Thank you to the author for addressing some of my concerns. but I still have some questions.
> >
> > 1. Regarding three-phase exact set sampling, I'm not sure if I understand it correctly. If IPC=200 and batch size=50, then the same group sampling will be performed 4 times, right? Is there any interaction between these four samplings, that is, will the previous sampling be used as input to affect the next sampling?
> >
> > 2. I still believe that the Spectral Set-Point Regulation is merely a metric referenced from NCFM, and only used in diffusion models. What I mean is that simple latent alignment from DM and the Wasserstein Metric as mentioned in [1] could easily replace it. The reason for my concern is that $V_{\mathrm{cpl}}$ is essentially a diversity loss. Using it alone shows limited improvement, or even worse performance (Table 5: 64.2% < 65.1%). This suggests that the performance gain mainly comes from the distribution metric. In addition, in Table 6, does IGD(-DEV)+SCG use $V_{\mathrm{reg}}$? If so, the comparison is not fair. I think it is necessary to compare IGD with IGD(-DEV)+$V_{\mathrm{cpl}}$, so that the effectiveness of group sampling can be properly demonstrated.
> >
> > 3. I also tried using only $V_{\mathrm{cpl}}$ in DiT, and indeed there was no clear performance improvement. Could you explain why this happens? Is it because the samples generated by DiT are already diverse enough? If so, does that mean the contribution of SCG mainly lies in introducing a distribution metric during the reverse process?
> >
> >
> >
> >
> >
> >  [1] Dataset Distillation via the Wasserstein Metric, ICCV, 2025.

---

> > > ### Author Response · Authors · 2026-04-01
> > >
> > > 1. Clarification of three-phase exact set sampling.
> > >
> > > When IPC=200, we do not perform four independent group samplings. Within the same reverse step t, we first process the 200 particles in four micro-batches only to collect predicted clean latents, descriptors, and base guidance. We then compute the set-level interaction once on the full 200-particle descriptor bank and write this precomputed controller back to the four micro-batches. Only the backbone execution is chunked for memory reasons; the set interaction remains global. These are four execution chunks of one set-coupled reverse step, not four separate samplings.
> > >
> > > 2. On whether spectral set-point regulation is merely a reused metric.
> > >
> > > We agree that the discrepancy itself is not the main novelty. Our contribution is to turn a set-level discrepancy objective into a low-cost closed-form reverse-time controller via projection, permutation-equivariant feedback, and lift back to latent space. The name spectral set-point regulation means that the controller tracks a target set statistic in the characteristic-function domain; it does not claim a new discrepancy metric by itself.
> > >
> > > Other set-level objectives can in principle replace the target objective inside this controller. What is compatible with our framework is the underlying objective, not necessarily the full method/pipeline from which it is used. An MMD-style objective can be inserted because it provides descriptor-space gradients on the current set state, whereas the full DM pipeline is a static outer-loop optimization over persistent synthetic variables and is therefore not directly comparable to our reverse-time controller design. So the comparison here is between objectives, not full pipelines. We also verified this in a descriptor-space pilot on ImageNette IPC10: replacing the ECF target with RBF-MMD still gives a valid closed-form controller with similar cost, but slightly lower ResNet-18 accuracy than our default ECF setting (64.6 vs. 65.8). In this sense, the core contribution is the controller design, not the objective alone.
> > >
> > > Concretely, for descriptor-space RBF-MMD,
> > > $\nabla_{\phi_k}V_MMD=(2/K^2)\sum_b\nabla_1 k(\phi_k,\phi_b)-(2/KN)\sum_i\nabla_1 k(\phi_k,r_i)$,
> > > which is closed-form; for the RBF kernel, $\nabla_1 k(x,y)=\sigma^{-2}k(x,y)(y-x)$. By contrast, for Wasserstein/OT objectives, although the position gradient is closed-form once the transport plan is fixed (e.g., $\nabla_{\phi_k}V_OT=2\sum_i \pi_{ki}(\phi_k-r_i)$), the transport plan $\pi$ itself must still be solved at each step. Thus ECF is not the only possible choice, but it is particularly practical here: it compresses the real-data reference into a fixed descriptor target, keeps per-step cost decoupled from the real-bank size, and aligns naturally with our current stability/tracking analysis.
> > >
> > > Regarding Table 6, you are correct that IGD(-DEV)+SCG uses the full SCG controller, i.e., both $V_{reg}$ and $V_{cpl}$. Thus Table 6 tests whether the full set-coupled controller can replace the sequential DEV mechanism in IGD, rather than isolating coupling alone. We agree that IGD(-DEV)+$V_{cpl}$ is the correct control for isolating “group coupling only”.
> > >
> > > 3. Why does $V_{cpl}$ alone show limited improvement?
> > >
> > > This is consistent with both our design and our theory. $V_{cpl}$  is not a standalone fidelity objective; it regularizes the joint trajectory once there exists a set-level target signal. $V_{reg}$ provides the tracking signal toward real-data statistics. In our ISS-style analysis, contraction comes from the regulation term, whereas the coupling term enters as a bounded stabilizing contribution. Therefore, when $V_{reg}$ is removed, there is no mechanism that forces the synthetic set statistics to approach the real target statistics.
> > >
> > > We do not interpret Table 5 as showing that $V_{cpl}$ is ineffective. $V_{cpl}$ alone improves over the baseline on ConvNet-6 and ResNet-18(58.6->60.0,61.5->62.2) and underperforms only on ResNetAP(65.1->64.2), so inferring ineffectiveness from a single evaluator is too strong. More importantly, full SCG vs. $V_{reg}$-only yields consistent gains on all three evaluators(61.2->62.0, 65.8->67.2, 63.4->64.1).
> > >
> > > This also explains why $\beta=0$ can still show nontrivial gains under IGD(-DEV) without contradiction. In IGD(-DEV)+$V_{cpl}$, the influence guidance is still present, so $V_{cpl}$ regularizes the remaining guided trajectory rather than operating in a vacuum. Our additional ImageNette/IPC10 ablation gives 58.2 / 66.2 / 64.0 for $\alpha=0$ and 62.2 / 61.6 / 64.8 for $\beta=0$, while ImageWoof/IPC10 gives 33.6/40.6/44.6 for $\alpha=0$ and 33.0 / 40.4 / 43.0 for $\beta=0$ on ConvNet / ResNetAP / ResNet18. Thus, under IGD, both terms can remain useful because they act on top of an existing influence-guided trajectory. We therefore do not interpret SCG as merely introducing a discrepancy objective into reverse diffusion; its gain comes from combining set-level tracking and set-level stabilization.

---

### Official Review · Reviewer_XipB · 2026-03-11

**Soundness:** 3
**Presentation:** 3
**Significance:** 3
**Originality:** 3
**Overall Recommendation:** 5
**Confidence:** 3

**Summary:**

This paper proposes Set-Coupled Guidance, a set-level sampling approach for diffusion-based dataset distillation, moving beyond the conventional per-image sampling paradigm. Set-Coupled Guidance consists of two components: spectral regulation and cooperative kernel coupling. Spectral regulation drives the synthetic sample set toward matching the distributional statistics of real data, while cooperative kernel coupling stabilizes the joint trajectory so that it does not break even when the guidance signal is perturbed by noise. The paper also provides theoretical analysis of the proposed method, and the experimental and ablation results support the effectiveness of the approach.

**Compliance With Llm Reviewing Policy:**

Affirmed.

**Final Justification:**

I would like to thank the authors for their comprehensive response to my previous concerns. After carefully reviewing the rebuttal and the additional experimental results provided, I find that the authors have addressed the core technical and empirical questions I raised. I will raise my score.

**Key Questions For Authors:**

1.	I understand that dataset distillation has a broader set of baselines[1,2] in the literature, but it seems that citations/comparison to some of these works are missing. It would also be helpful to include experimental results on CIFAR-100.
2.	In my view, set-level sampling might be more effective when IPC is large. Do you provide any analysis of how the benefit of set-level sampling changes as a function of IPC?
3.	I noticed qualitative results in Figure 5 of the appendix, but they only show a single class; I would like to see more diverse examples. Also, is there a way to compare the effects of set-level sampling versus image-level sampling directly using the final generated images (e.g., via set-level metrics)?
4.	(Minor) Can set-coupled guidance be applied in class-imbalanced settings as well?

[1] Enhancing Dataset Distillation via Non-Critical Region Refinement, CVPR 2025

[2] Multimodal Dataset Distillation Made Simple by Prototype-Guided Data Synthesis, ICLR 2026

**Limitations:**

What is the main limitation of the proposed method?

**Strengths And Weaknesses:**

**Strengths**

* The paper is well written and easy to follow, and the motivation for set-coupled guidance is convincing.

* Beyond the theoretical analysis, the extensive experiments provide strong evidence supporting the effectiveness of the proposed approach.

* Set-coupled guidance highlights an aspect that has been largely overlooked in dataset distillation, and it consistently improves performance when plugged into existing DD methods with minimal overhead.

**Weaknesses**

* The descriptor projection and lift use a fixed per-class PCA basis fit on real latents; the effect of PCA rank limitations and approximation error on guidance optimality is not fully characterized beyond boundedness assumptions.

* The theoretical analysis relies on assumptions such as bounded perturbations and local well-conditioning of the ECF map. Although these assumptions may be reasonable, it is not clear how often they hold in practice or how they could be verified for a given dataset or model.

* Missing comparisons in the main paper with several relevant recent guidance/distillation baselines (e.g., OT-GDD, IGD in main tables, DAP/IGDS), which would better establish relative gains from set-level coupling.

---

> ### Author Rebuttal · Authors · 2026-03-30
>
> Response to Reviewer XipB
>
> Thank you for your constructive suggestions.
>
> **Q1: Missing comparisons/citations to some recent DD works [1], [2], and CIFAR-100 results.**
>
> **A1:** In the revision, we will add the missing citations and discussion of [1] and [2]. We also ran an additional CIFAR-100 experiment under a hard-label-only `convnet5` evaluation setting (`SGD  + basic crop/flip + 200 epochs`) using a CIFAR-100 diffusion checkpoint trained by us:
>
> | Method | IPC10 | IPC20 | IPC50 |
> | :--- | ---: | ---: | ---: |
> | random real subset | 15.2 | 21.9 | 32.7 |
> | MGD$^3$ | 15.2 | 22.3 | 32.6 |
> | MGD$^3$+SCG | **16.1** | **23.2** | **35.7** |
>
> SCG consistently improves MGD$^3$ on this CIFAR-100 setting, and we will include these results in the revision.
>
> **Q2: Analysis of how the benefit of set-level sampling changes as a function of IPC.**
>
> **A2:** Yes. Combining the ImageNette ResNet-18 results already reported in the paper (IPC=10/20/50) with our new higher-IPC runs gives:
>
> | IPC | MGD$^3$ | MGD$^3$+SCG | Improvement |
> | :--- | :---: | :---: | :---: |
> | 5 | 53.4% | **57.2%** | +3.8% |
> | 10 | 61.5% | **64.1%** | +2.6% |
> | 20 | 69.1% | **73.0%** | +3.9% |
> | 50 | 78.1% | **82.6%** | +4.5% |
> | 100 | 83.0% | **84.4%** | +1.4% |
> | 200 | 87.6% | **88.2%** | +0.6% |
>
> Within the currently observed range, the gain is not monotone in IPC: it is strongest at medium IPC (20--50) and then decreases at IPC=100/200. A simple statistical interpretation is that as IPC grows, the empirical set already better approximates the target distribution, and the final evaluator accuracy also moves closer to its saturation ceiling; therefore the marginal benefit of explicit set-level coordination should diminish. We will add this trend analysis in the revision.
>
> **Q3: More diverse qualitative examples and direct comparison using set-level metrics.**
>
> **A3:** We will expand the appendix in the revised manuscript to include more diverse qualitative examples from multiple classes. As an additional supplement during the rebuttal period, we also provide more diverse qualitative examples in our anonymous repository: https://anonymous.4open.science/r/pictures-6CCE
>
> Regarding set-level metrics, Appendix C.1 already provides selected-class RBF-MMD examples at IPC=10. We additionally computed 10-class average RBF-MMD at IPC=50: on ImageNette, MGD$^3$ decreases from $0.00983 \pm 0.00282$ to $0.00958 \pm 0.00156$; on ImageWoof, it decreases from $0.01129 \pm 0.00310$ to $0.00988 \pm 0.00155$. We will add these dataset-level set metrics in the revision.
>
> **Q4: Can set-coupled guidance be applied in class-imbalanced settings as well?**
>
> **A4:** In the setting studied here, where SCG is plugged into a pretrained diffusion prior, it should remain applicable under class imbalance because the imbalance only changes the per-class real descriptors/statistics used to compute the guidance terms. What we do not claim here is training a diffusion model from scratch under class imbalance; that lies outside the scope of this paper. As a brief sanity check, we ran a long-tail ImageNette experiment in which the real data follow an imbalance factor of 10 (largest/smallest class count ratio about 10:1), while the distilled output remains balanced at IPC=10 per class. Under ResNet-18 evaluation, MGD$^3$ improves from 62.4 to 65.0 after adding SCG, supporting the claim that SCG remains usable when the real descriptor bank is class-imbalanced. We will add this clarification and the preliminary result in the revision.
>
> **Response to weaknesses:**
>
> *   **PCA rank limitations:** Our PCA-rank ablation shows that reconstruction MSE decreases monotonically with $D$, while downstream accuracy remains stable at 65.4%--66.2%; the detailed results are provided below in our response to Reviewer fzoG.
> *   **Theoretical assumptions:** As detailed in our response to Reviewer oium, we directly verified the practical reasonableness of these assumptions on both ImageNette and ImageWoof. In particular, local well-conditioning holds throughout the active window in our diagnostics, and the measured backbone/observation mismatch remains small, with the mismatch error normalized by the noise scale $\sigma_t$ decreasing across the active window.
> *   **Comparison coverage:** We agree that the comparison coverage in the main paper should be broader. IGD results are already included in Appendix B.3: across ImageNette/ImageWoof and IPC=10/20, IGD(-DEV)+SCG recovers and often matches or exceeds full IGD while avoiding the sequential DEV-based generation order. Because the main paper is already space-constrained, we kept the detailed IGD comparison in the appendix and will improve its visibility in the revision. For `OT-GDD` and `DAP/IGDS`, empirical reproduction during the rebuttal window was not feasible because public implementations were not available to us, so in the revision we will at least add the missing citations and discussion.

---

> > ### Author Rebuttal · Reviewer_XipB · 2026-04-03
> >
> > I find that the authors have addressed the core technical and empirical questions I raised.

---

> > > ### Author Response · Authors · 2026-04-03
> > >
> > > Thank you for the follow-up and for revisiting your score. We sincerely appreciate your careful review and are glad that our rebuttal addressed the core technical and empirical concerns you raised.

---

### Official Review · Reviewer_oium · 2026-03-13

**Soundness:** 2
**Presentation:** 3
**Significance:** 2
**Originality:** 2
**Overall Recommendation:** 3
**Confidence:** 3

**Summary:**

The paper is essentially trying to do better dataset distillation (DD). DD requires us to create a dataset that preserves key properties of the original dataset from a ML point of view, however, prior approaches in DD seem to focus mostly on transforming individual elements, whereas the goal is to produce a new dataset, i.e. a set. Hence, the paper introduces internal coupling terms that are basically trying to ensure that this new set as a whole reflects the original term better. They test their method as an add-on to existing methods like CaO2.

**Compliance With Llm Reviewing Policy:**

Affirmed.

**Final Justification:**

Resnet-18 is too small for 2026, but the authors managed to provide at least some larger models in the rebuttal, causing me to change the score from 2 to 3.

**Key Questions For Authors:**

Can the authors explain why the empirical evaluation is remarkably small-scale for 2026 ? Privacy is mostly an issue for larger-scale models in production, not Resnet-18 scale models.

**Limitations:**

yes

**Strengths And Weaknesses:**

The idea is well written and visualized. I was able to follow the idea despite low exposure to diffusion models, etc. and the method seems to work as an add-on to various existing methods to always improve them, which is often a key requirement, as we do not know which new method will emerge to beat existing methods soon.

Weaknesses : The empirical evaluation seems very weak. Resnet-18 etc. are very tiny models by 2026 standards. I also felt that the theory is longer than it needs to be, and makes too strong assumptions re : stability etc. that make the final derivations unsurprising.

---

> ### Author Rebuttal · Authors · 2026-03-30
>
> Response to Reviewer oium
>
> Thank you for the review. We respond to your two main concerns:
>
> **Q1: The empirical evaluation is remarkably small-scale (e.g., ResNet-18). Why not use larger models?**
>
> **A1:** We agree that larger evaluators matter. Our submission followed the standard DD evaluation protocol used in recent diffusion-based DD work, including MGD$^3$: *Mode-Guided Dataset Distillation using Diffusion Models* (ICML 2025) and *Diffusion Models as Dataset Distillation Priors* (ICLR 2026). In this literature, standard DD evaluators remain the dominant reporting protocol, while larger-model evaluation is typically included as a complementary generalization check. While these evaluators are smaller than production-scale models, in DD they mainly test whether the distilled set transfers across architectures under a fixed protocol; larger-model evaluation is therefore an additional stress test, not the only relevant measure.
>
> To address your concern directly, we added IPC=50 cross-architecture evaluations on substantially larger backbones for both ImageNette and ImageWoof:
>
> | Architecture | ImageNette (MGD$^3$ → +SCG) | ImageWoof (MGD$^3$ → +SCG) |
> | :--- | :--- | :--- |
> | ResNet-50 | 73.8% → **75.0%** | 43.6% → **46.0%** |
> | ResNet-101 | 70.4% → **74.0%** | 40.0% → **41.2%** |
> | ViT-Small | 59.2% → **59.4%** | 34.8% → **37.2%** |
> | ViT-Base | 62.0% → **65.8%** | 39.2% → **40.2%** |
> | ViT-Large | 64.2% → **67.8%** | 38.2% → **39.2%** |
> | Swin-Base | 69.4% → **70.2%** | 38.6% → **40.2%** |
>
> These results show that SCG is not limited to ConvNet/ResNet-18-scale evaluators. Due to rebuttal compute limits, we could not expand these runs to more datasets or IPC settings; we will include more such results in the final appendix.
>
> **Q2: The theoretical assumptions (bounded perturbations, local well-conditioning) seem too strong.**
>
> **A2:** We discuss them proposition by proposition.
>
> - **Proposition 1 (Lyapunov decrease).** It assumes bounded effective disturbance together with standard smoothness on the descriptor manifold. In the appendix, we separate three practical contributors to this disturbance: backbone/observation mismatch, lift approximation, and implementation-side stabilizers. For backbone/observation mismatch, we use a first-order proxy: locally linearize the descriptor map at the current predicted clean latent, apply the resulting Jacobian-vector product to the SCG latent update, and compare it with the actual one-step descriptor change (ImageNette / ImageWoof, IPC=10, $D=64$, $K_f=32$). The median absolute mismatch is $1.044\times 10^{-3}$ / $1.077\times 10^{-3}$, and the mismatch normalized by $\sigma_t$ decreases from $5.73\times 10^{-3}$ to $3.65\times 10^{-4}$ on ImageNette and from $5.15\times 10^{-3}$ to $4.29\times 10^{-4}$ on ImageWoof across the active window. For lift approximation, PCA diagnostics on ImageNette show that reconstruction MSE and residual error decrease monotonically with $D$, while the lifted update stays directionally aligned with a high-rank reference. This supports bounded lift error in practice, without claiming it vanishes. Separately, under our default settings the realized controller step matches the ideal projected step throughout the active window, so the practical `nan_to_num` / RMS-clipping stabilizers add no measurable extra disturbance in the main experiments. We therefore do not claim exact one-step descriptor prediction; rather, the evidence supports that the dominant mismatch remains bounded and shrinks relative to $\sigma_t$, while lift error and implementation-side stabilizers stay controlled in the default regime.
>
> - **Proposition 2 (ISS tracking).** It additionally assumes local well-conditioning of the ECF map, i.e., $J_h J_h^T \succeq \mu I$, together with annealed coupling gain. This assumption is directly testable. On real MGD$^3$+SCG trajectories (ImageNette / ImageWoof, IPC=10, $D=64$, $K_f=32$, 50 diffusion steps), we evaluated all $21 \times 10 = 210$ class-step states in the regulation-active window for each dataset and found that the well-conditioning fraction, defined as $\lambda_{\min}(J_h J_h^T) > 10^{-4}$, is 1.000 on both datasets. The median $\lambda_{\min}$ is 6.456 on ImageNette and 5.983 on ImageWoof; even at the earliest active step ($t/T=0.7$), the median $\lambda_{\min}$ remains 1.898 and 2.144, respectively. The verification procedure is therefore direct for any dataset/model pair: compute $J_h J_h^T$ on active class-step states and report the fraction above a threshold such as $10^{-4}$.
>
> - **Proposition 3 (anti-collapse).** It assumes a non-degenerate real class distribution and at least one informative sampled anchor. This is mild: any non-point-mass real distribution admits such a nonzero anchor, and random anchor sampling captures one with high probability as $K_f$ grows.
>
> We will add this proposition-by-proposition clarification and the above diagnostics to the revised manuscript.

---

> > ### Author Rebuttal · Reviewer_oium · 2026-04-03
> >
> > Okay, this is much better ! Thank you for running experiments on going beyond Resnet-18. Given the larger models, I will be changing my score. Please make them part of the main paper going forward.

---

> > > ### Author Response · Authors · 2026-04-04
> > >
> > > Thank you for the follow-up and for revisiting your score. We are glad that the additional larger-model evaluations addressed the concern. We confirm that these results will be included in the main paper in the final revision, not left only in the rebuttal or appendix.

---

### Decision · Program_Chairs · 2026-04-30

**Decision:**

Accept (regular)

**Comment:**

This paper addresses an important limitation of diffusion-based dataset distillation: although distilled data are evaluated as a set, most existing methods still update samples independently. To address this mismatch, the paper proposes Set-Coupled Guidance (SCG), a plug-and-play controller that injects set-symmetric feedback into each diffusion step, enabling group-wise optimization of the in-class synthetic set.

Reviewers found the motivation clear and the method technically distinctive. In particular, the paper’s strengths include its closed-form, set-symmetric guidance law and its theoretical analysis of convergence, stability, and anti-collapse behavior.

The main concerns are empirical. As Reviewer oium noted, the original evaluation was conducted at relatively small model scales. In the rebuttal, the authors provided additional larger-model results, which should be incorporated into the final version more fully. Reviewer XipB and fzoG also pointed out that experiments at larger IPC values would better align with the paper’s central motivation of improving set-level diversity and coverage at scale. While the rebuttal extended results up to IPC 200 on ImageNette with ResNet-18, the gains were not fully consistent across IPC values: improvements were strongest at moderate IPC and diminished at larger IPC, raising some concern about how broadly the benefits scale in the regime where set-level coordination should matter most.

Despite these limitations, the overall assessment is positive. The paper identifies an important methodological gap and proposes a principled, technically distinctive solution. I therefore recommend acceptance. I encourage the authors to incorporate the additional rebuttal results in the final version.